# A Global ArUco-Based Lidar Navigation System for UAV Navigation in GNSS-Denied Environments

**Ziyi Qiu** [1,2], **Defu Lin** [1,2], **Ren Jin** [1,2,*], **Junning Lv** [1,2] **and Zhangxiong Zheng** [1,2]

1   School of Aerospace Engineering, Beijing Institute of Technology, Beijing 100081, China
2   Beijing Key Laboratory of UAV Autonomous Control, Beijing Institute of Technology, Beijing 100081, China
*   Correspondence: renjin@bit.edu.cn; Tel.: +86-132-2012-0166

**Abstract:** With the continuous expansion of the application field of UAV intelligent systems to GNSS-denied environments, the existing navigation system can hardly meet low cost, high precision, and high robustness in such conditions. Most navigation systems used in GNSS-denied environments give up the connection between the map frame and the actual world frame, making them impossible to apply in practice. Therefore, this paper proposes a Lidar navigation system based on global ArUco, which is widely used in large-scale known GNSS-denied scenarios for UAVs. The system jointly optimizes the Lidar, inertial measurement unit, and global ArUco information by factor graph and outputs the pose in the real-world frame. The system includes a method to update the global ArUco confidence with sampling, improving accuracy while using the pose solved from the global ArUco. The system uses the global ArUco to maintain navigation when Lidar is degraded. The system also has a loop closure determination part based on ArUco, which reduces the consumption of computing resources. The navigation system has been tested in the dry coal shed of a thermal power plant using a UAV platform. Experiments demonstrate that the system can achieve global, accurate, and robust pose estimation in large-scale, complex GNSS-denied environments.

**Keywords:** UAV; GNSS-denied environments; Lidar; navigation system; factor graph optimization

## 1. Introduction

In the UAV system [1], the navigation system is responsible for providing carrier location information [2], which guarantees the UAV to complete the task safely and accurately [3]. Under the development trend of UAV system intelligence [4], traditional GNSS navigation systems [5,6] have been unable to complete the application of UAV systems in GNSS-denied environments in recent years [7,8]. The demand for a high-precision GNSS-denied navigation system is urgent in such aspects as the mine UAV exploration system, thermal power plant dry coal shed UAV square measurement system, Power Plant Turbine workshop daily UAV inspection system, high-speed railway platform truss structure anti-corrosion coating UAV inspection system, etc.

Here, I would like to mainly introduce one of the working conditions that have been actually applied in this paper, which is also the scene used in this test the square measurement system of UAV in a dry coal shed of the thermal power plant. The system's primary function is to correct the total amount of coal in and out of the thermal power plant in about 2 to 3 days to better allocate the fuel in the future. The crucial part of the system is measuring the coal volume in the dry coal shed. The traditional manual operation generally scans the coal stacked in the coal shed by climbing to a few points people can reach through the handheld laser radar. The sampling angle is limited, the blind area is large, and the sampling personnel are exposed to the risk of falling. Therefore, an unmanned system is required to replace manual operation. However, due to the complex natural environment in the dry coal shed, if cameras, slide rail cameras, slide rail Lidar, or other detection equipment are installed around the shed, the construction and installation

cost will be very high, and the daily cleaning and maintenance of the equipment will not be completed. Similarly, the electromagnetic environment in the coal shed is also very complex, which challenges the use of more intelligent systems such as UAVs. For example, the automatic flight of the UAV system in this environment requires accurate positioning. Still, the complex electromagnetic environment will cause the deviation of the magnetic compass in the traditional navigation system. It will also affect the positioning accuracy of external navigation such as UWB. The thermal power plant dry coal shed UAV square measurement system diagram is shown in Figure 1. It can be observed that a high-precision navigation system is vital in a UAV system when GNSS is denied.

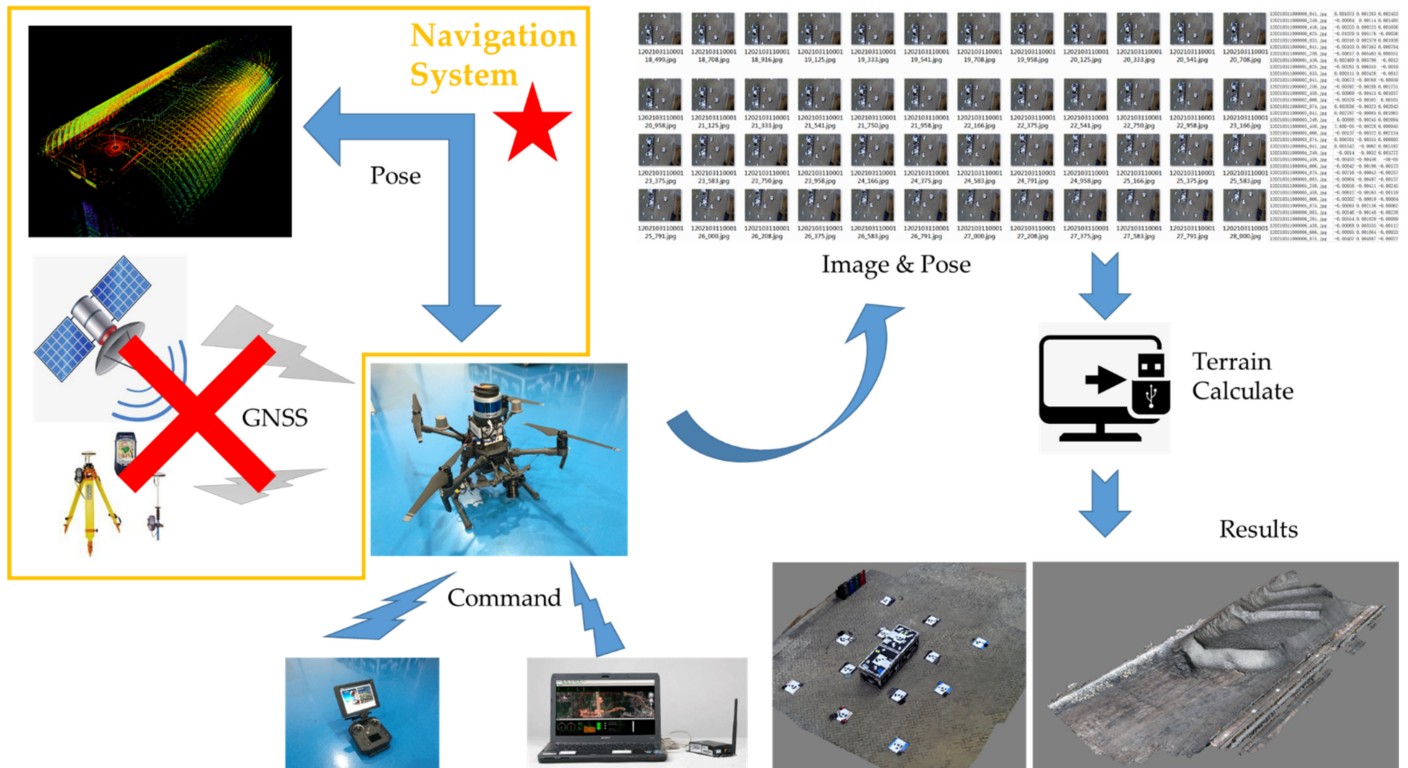

**Figure 1.** The thermal power plant dry coal shed UAV square measurement system.

Currently, navigation in the GNSS denial environment mainly relies on the UWB, vision sensor, Lidar, and inertial unit. External navigation generally includes UWB and Vicon systems. Among them, the UWB positioning preset hardware cost is high, the blind area is large, the maintenance is complex, and the accuracy is easily affected by dust and metals in the environment [9]. Before using the system described in this paper, the Jiangxi Fengcheng thermal power plant installed six UWB base stations in the coal shed. The cost of setting up one UWB only for construction, installation, and construction cooperation was about 30 thousand CHF. However, due to blind areas and electromagnetic interference, the UAV navigation could not be completed. Vicon system uses multiple cameras pre-installed and calibrated with high precision to locate the target, which also has the disadvantages of a large blind area and high cost. For example, the Vicon system of the Beijing Key Laboratory of UAV Autonomous Control of Beijing University of technology actually occupies a range of 9.6 m × 7.8 m × 2.8 m. However, the available high-precision area is only about 6 m × 5 m × 1.8 m in the middle. The cost of the system is nearly 300 thousand CHF. Thus, Vicon system is not feasible to use in large-scale production and life. Visual sensors are mainly used with the inertial unit as a visual inertial navigation system, such as ORB-SLAM2 [10], VINS-Mono [11], etc. This type of visual inertial navigation system is susceptible to changes in light in the working environment and has strict requirements for the features of the environment. It can hardly work properly when light changes or visual features are not

obvious. Lidar is mainly combined with the inertial unit to form a Lidar inertial navigation system, such as LOAM [12], LeGO-LOAM [13], FAST-LIO [14], LIOM [15], etc. In recent years, some multi-sensor navigation systems have fused more sensors. Such systems integrate visual sensors, Lidars, inertial units, GNSS sensors, etc., in different ways, such as LIO-SAM [16], LVI-SAM [17], etc. This navigation system complements the advantages and disadvantages of their respective sensors but occupies more computing resources.

The above traditional vision, laser, and inertial navigation systems can not accurately pose in the world frame that is fixedly connected with the actual scene in the GNSS-denied environment. It is affected by each system restart and can only provide the relative pose based on the pose when the system is started. It cannot correct the cumulative error through the actual pose information, so those navigation systems cannot guarantee the positioning accuracy in practical applications, especially in large-scene applications. Moreover, in the case of failure and degradation, the above algorithm cannot restore navigation, seriously affecting operation safety in practical applications. Therefore, this paper proposes a global ArUco-based Lidar navigation system suitable for UAVs in a GNSS-denial environment.

ArUco, the Augmented Reality University of Cordoba, was originally proposed by Garrido-Jurado, S. and others in reference [18] published in 2014. Here is an example of ArUco marks shown in Figure 2. The ArUco mark is a square with a black background. The interior is marked with a white pattern to indicate the mark's uniqueness. It can be arbitrarily modified to the appropriate size according to the requirements of the application scenario. While using, the camera is used to collect images, and the images are detected by the onboard computer. If the image contains ArUco, the relative position relationship between ArUco and the camera can be obtained through simple calculations.

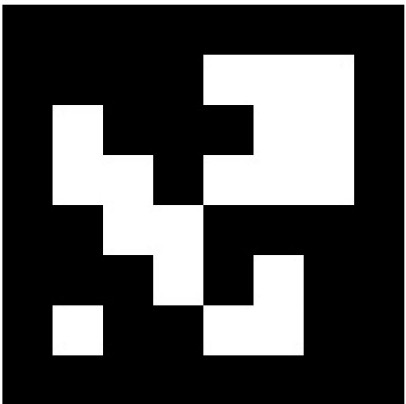

**Figure 2.** The example of ArUco markers.

The global ArUco described in this paper are ArUco markers fixed to the earth. It is placed or printed in the designated position in the coal shed. Like the building, it is fixed. It has unique longitude, latitude, and altitude coordinates in the ECEF system. It will not change with the position and attitude of a restart of the navigation system every time. For the convenience of calculation, it will be converted to a metric system according to the application scenario. Still, the nature of its fixed connection with the geodetic coordinate system will not be changed.

In the automotive field, there have been systems that use ArUco for auxiliary navigation [19]. However, these systems need ArUco to participate in initial navigation, which has many limitations. Moreover, these systems only place ArUcos in the places concerned by tasks such as parking spaces. Such ArUcos are mainly used to assist in setting specific tasks and only ensure the accuracy near specific tasks. These navigation systems cannot perform global corrections to the pose for large-scale and multi-dimensional motion. In contrast, the navigation system proposed in this paper also uses ArUco, and can more flexibly correct the global pose in large-scale and multi-dimensional motion.

The main work of this paper is as follows:

✦ The factor graph structure of the Lidar navigation system based on global ArUco is constructed, which can fuse the sensor data globally. The accuracy and robustness of the navigation system can be significantly improved by combining the processing method proposed in this paper when the Lidar motion solution is degraded.

✦ A global ArUco factor is constructed, which can update confidence accurately according to sampling. This factor participates in the optimization as a priori of the state in the factor graph, which ensures that the navigation system can work in the geodetic coordinate system fixed with the actual scene and corrects the error of the navigation system according to the actual scene. Compared with traditional vision, it improves the accuracy of the navigation system and reduces the use of computing resources, and enhances real-time performance.

✦ A loopback determination method based on global ArUco is constructed, making loopback detection more accurate and efficient.

✦ The navigation system described in this paper is tested using the UAV platform in the dry coal shed of thermal power plants, one of the practical application scenarios, and compared with other Lidar algorithms.

The main contents of this paper are as follows. The second chapter introduces the related work of others. The third chapter introduces the algorithm framework of the navigation system and the factors of the factor graph. The fourth chapter introduces the calibration method of noise covariance in the global ArUco real-time measurement, the navigation system experiment in an actual working condition and the navigation system accuracy test. The fifth chapter is the conclusion.

## 2. Related Work

In recent years, the visual navigation system mainly includes the following work. ORB-SLAM2 [10] proposed a new tight coupling vision inertial navigation system, adding loop closure detection to correct the drift of the navigation system. VINS-Mono [11] used nonlinear optimization to fuse pre-integrated IMU measurements and feature observations. It used a tightly coupled formulation combination with a loop closure detection module to save the cost of computation resources. Moreover, it performed factor graph optimization to enforce global consistency. The navigation system described in this paper mainly uses Lidar and only uses vision to identify ArUco. The computing resources of the visual part are much lower than the above-mentioned visual inertial navigation system.

The main algorithms of the Lidar navigation system mainly include the following work. LOAM [12] first proposed a straightforward point cloud compensation method and planar feature and edge feature extraction method. It used distances from a point to a plane and a point to a line as a cost function to match the frame and estimate the motion. It also proposed a back-end pair optimization algorithm that simultaneously outputs the pose in high-frequency low-precision and low-frequency high-precision. The idea of the algorithm is widely used for reference and improved by other Lidar navigation systems. However, the algorithm cannot deal with large-scale rotation transformation. LeGO-LOAM [13] divides the point cloud into the ground and other point clouds. Firstly, Z, roll, and pitch are optimized through the segmented ground point cloud, and then x, y, and yaw are optimized through other point clouds. The six-dimensional optimization is simplified into two three-dimensional optimizations, reducing computational complexity. The algorithm also eliminates the dynamic point noise when extracting features and classifies the features during matching, which further reduces the occupation of computing resources. However, the algorithm's advantages cannot be realized because the Lidar cannot illuminate the ground during the operation of a UAV system. The above navigation systems all rely on Lidar, and the navigation system will fail when the Lidar motion solution degrades. The navigation system described in this paper enhances the robustness of the Lidar motion solution when it degrades through global ArUco.

The multi-sensor fusion navigation system mainly includes the following work. LIO-SAM [16] removes frame-to-frame matching in traditional Lidar point cloud matching

and only matches keyframes, which reduces the use of computing resources. Add GNSS sensors to the navigation system and use factor graph optimization to enhance the fusion of individual sensor data. The algorithm can exert its advantages in the GNSS environment but cannot make global corrections to the navigation results in the GNSS-denied environment.

## 3. ArUco-Based Lidar Navigation System for UAVs in GNSS-Denial Environment

For the convenience of later description, an operator of transforming the Euler angle into a rotation matrix is defined here.

Define the Euler angle as:

$$\boldsymbol{\Omega} = \begin{pmatrix} r & p & y \end{pmatrix}^T$$

The operator is shown as:

$$\Re(\boldsymbol{\Omega}) = \begin{pmatrix} \cos y \cos p & \cos y \sin p \sin r - \sin y \cos r & \cos y \sin p \cos r + \sin y \sin r \\ \sin y \cos p & \sin y \sin p \sin r + \cos y \cos r & \sin y \sin p \cos r - \cos y \sin r \\ -\sin p & \cos p \sin r & \cos p \cos r \end{pmatrix}$$

### 3.1. System Overview

The navigation system described in this paper uses surround Lidar, IMU, gimbal, camera, and global ArUco markers as system information sources. The surround Lidar, IMU, and gimbal are fixedly connected with the UAV, and their relative pose relationship is calibrated. The calibration method is the same as that in reference [20]. The gimbal can output the pose relationship between the camera and the UAV. The global ArUco marks are placed in a known GNSS-denied environment scene, and its absolute pose in the scene is known.

In order to consider the accuracy of the navigation system globally, optimization rather than filtering is selected in the main framework of sensor observation fusion. The main body of the navigation system designed in this paper is to solve a nonlinear optimization problem for state estimation, use sensor observations to solve the maximum posterior probability estimation of the state, and use factor graphs to express the relationship between them [21], and use iSAM2 to solve [22]. In the optimization, we consider the state of the UAV at time k as:

$$\mathbf{X_k} = \begin{pmatrix} \mathbf{p_k} & \mathbf{v_k} & \mathbf{R_k} & \mathbf{b_{ak}} & \mathbf{b_{gk}} \end{pmatrix}^\mathbf{T}$$

where $\mathbf{p_k}$ is the translation vector of the UAV in the world frame, $\mathbf{v_k}$ is the velocity vector of the UAV in the world frame, $\mathbf{R_k} \in SO(3)$ is the rotation matrix of the UAV in the world frame, and $\mathbf{b_{ak}}$ and $\mathbf{b_{gk}}$ are the bias of accelerometer and gyro in IMU, respectively.

The navigation system adds the global ArUco factor, IMU pre-integration factor, Lidar factor, and global ArUco loop closure factor in the factor graph. The structure of the factor graph is shown in Figure 3. At time k, take the global ArUco factor as the initial optimization value. Furthermore, add the IMU pre-integration factor calculated by IMU pre-integration between time k-1 and k and the Lidar factor calculated by Lidar motion estimation between time k-1 and k. Moreover, determine the loopback through the global ArUco. If the loopback occurs at time j, continue to add the loopback factors between time j and time k to the factor graph.

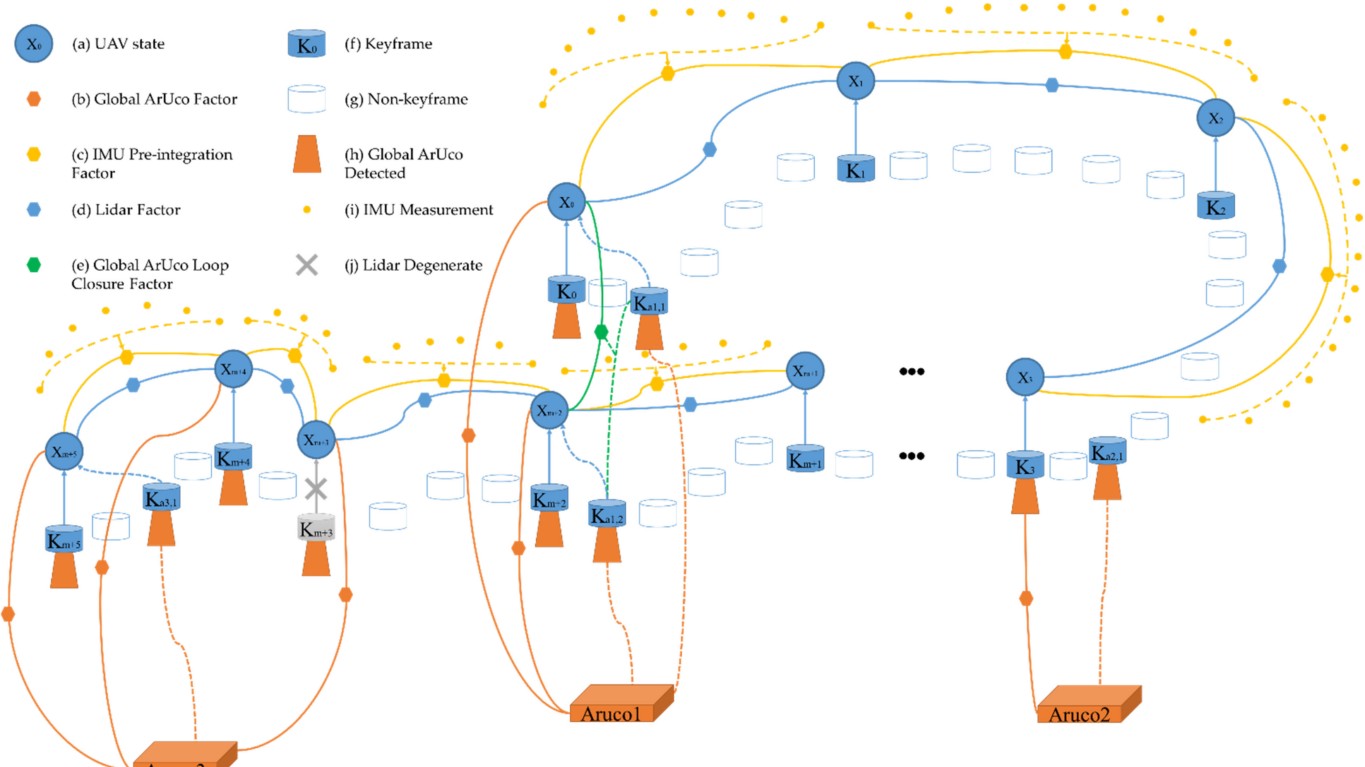

**Figure 3.** The structure of the factor graph. The system receives input from a camera, an IMU, and a surround Lidar. The factor graph is constructed by four types of factors: (b) global ArUco factor, (c) IMU pre-integration factor, (d) Lidar factor, and (e) global ArUco loop closure factor. The generation of these factors is discussed in the following sections, respectively.

### 3.2. Global ArUco Factor

In order to avoid the uncorrectable error caused by the navigation system only using the IMU and Lidar data for pose calculation for a long time, the global ArUco factor is added to the factor graph in this navigation system. The global ArUco factor can fixedly connect and align the map frame, used to calculate IMU and Lidar, with the world frame. Moreover, it can also correct the position and attitude error generated by the calculation of IMU and Lidar through the global pose. What is more, it can be used as the information source to maintain the regular operation of the navigation system when the matching of the Lidar point cloud is degraded.

The global ArUco factor comprises ArUco detection, global pose calculation, covariance comparison, and EKF sensor fusion process.

The global ArUco is placed in the known task scenario as required in advance. When the navigation system works, the image is captured by the camera. First, the system will grayscale and binarize the image. Secondly, the system will detect quadrilaterals, remove similar quadrilaterals, obtain bit assignment from the image, and obtain the coordinates of the four corners of ArUco in the image frame and correct them. Finally, the system will solve the rotation matrix and translation vector of the ArUco relative to the camera frame by using the PNP method [18]. The ArUco detection and solution function library used in this paper are shown in references [23].

The rotation vector and translation vector of the ArUco numbered $i$ relative to the camera frame are $\mathbf{r}_i$ and $\mathbf{t}_i$, respectively. The navigation system calculates the global pose $\mathbf{T}^{\mathbf{lidar}}_{\mathbf{world}i}$ of the Lidar relative to the actual world frame according to the Euler angles $\begin{pmatrix} \varphi_{gi} & \theta_{gi} & \phi_{gi} \end{pmatrix}^T$ of the gimbal, the position relationship $\begin{pmatrix} x_g & y_g & z_g \end{pmatrix}^T$ between the Lidar and the gimbal obtained from the calibration, and the pose $\begin{pmatrix} x_i & y_i & z_i & \varphi_i & \theta_i & \phi_i \end{pmatrix}^T$ of the

ArUco numbered $i$ in the known scene. The specific form of $\mathbf{T}^{\mathbf{lidar}}_{\mathbf{world}i}$ is shown in Equation (1). Below, $\mathbf{T} \in SE(3)$.

$$\mathbf{T}^{\mathbf{lidar}}_{\mathbf{world}i} = \mathbf{T}^{\mathbf{lidar}}_{\mathbf{gimble}} \mathbf{T}^{\mathbf{gimble}}_{\mathbf{camera}i} \mathbf{T}^{\mathbf{camera}}_{\mathbf{aruco}\,i} \mathbf{T}^{\mathbf{aruco}}_{\mathbf{world}i} \tag{1}$$

where $\mathbf{T}^{\mathbf{aruco}}_{\mathbf{world}i}$ is the transformation matrix of the pose of the ArUco numbered $i$ frame in the world frame, $\mathbf{T}^{\mathbf{camera}}_{\mathbf{aruco}\,i}$ is the transformation matrix of the pose of the camera frame in the ArUco numbered $i$ frame, $\mathbf{T}^{\mathbf{gimble}}_{\mathbf{camera}i} = \mathbf{T}^{\mathbf{camera}\,-1}_{\mathbf{gimble}i}$, $\mathbf{T}^{\mathbf{gimble}}_{\mathbf{camera}i}$ is the inverse matrix of the pose transformation matrix of the camera in the gimbal frame when the ArUco numbered $i$ is detected, $\mathbf{T}^{\mathbf{lidar}}_{\mathbf{gimble}}$ is the position transformation matrix of the Lidar in the gimbal frame obtained from calibration.

The relationship between $\mathbf{T}^{\mathbf{aruco}}_{\mathbf{world}i}$ and $\begin{pmatrix} x_i & y_i & z_i & \varphi_i & \theta_i & \phi_i \end{pmatrix}^T$ is shown in Equation (2).

$$\mathbf{T}^{\mathbf{aruco}}_{\mathbf{world}i} = \begin{pmatrix} \mathbf{R}^{\mathbf{aruco}}_{\mathbf{world}i} & \mathbf{t}^{\mathbf{aruco}}_{\mathbf{world}i} \\ 0 & 1 \end{pmatrix} \tag{2}$$

where $\mathbf{R}^{\mathbf{aruco}}_{\mathbf{world}i}$ is the rotation matrix of the ArUco numbered $i$ to the world frame, and the specific expression is shown in Equation (3). $\mathbf{t}^{\mathbf{aruco}}_{\mathbf{world}i}$ is the translation vector of the ArUco numbered $i$ to the world frame, and the specific expression is shown in Equation (4).

$$\mathbf{R}^{\mathbf{aruco}}_{\mathbf{world}i} = \Re \begin{pmatrix} \varphi_i \\ \theta_i \\ \phi_i \end{pmatrix} \tag{3}$$

$$\mathbf{t}^{\mathbf{aruco}}_{\mathbf{world}i} = \begin{pmatrix} x_i & y_i & z_i \end{pmatrix} \tag{4}$$

The relationship between $\mathbf{T}^{\mathbf{camera}}_{\mathbf{aruco}\,i}$, $\mathbf{r}_i$, and $\mathbf{t}_i$ is shown in Equations (5) and (6).

$$\mathbf{T}^{\mathbf{camera}}_{\mathbf{aruco}\,i} = \mathbf{T}^{\mathbf{aruco}\,-1}_{\mathbf{camera}i} \tag{5}$$

$$\mathbf{T}^{\mathbf{aruco}}_{\mathbf{camera}i} = \begin{pmatrix} \mathbf{R}^{\mathbf{aruco}}_{\mathbf{camera}i} & \mathbf{t}_i \\ 0 & 1 \end{pmatrix} \tag{6}$$

where $\mathbf{R}^{\mathbf{aruco}}_{\mathbf{camera}i}$ is the rotation matrix of the ArUco numbered $i$ in the camera frame, as shown in Equation (7).

$$\mathbf{R}^{\mathbf{aruco}}_{\mathbf{camera}i} = \cos\alpha_i \mathbf{I} + (1 - \cos\alpha_i)\mathbf{r}_i\mathbf{r}_i^{\mathbf{T}} + \frac{\sin\alpha_i}{\alpha_i} \begin{pmatrix} 0 & -r_{iz} & r_{iy} \\ r_{iz} & 0 & -r_{ix} \\ -r_{iy} & r_{ix} & 0 \end{pmatrix} \tag{7}$$

where in Equation (7), $r_{ix}$, $r_{iy}$, and $r_{iz}$ are the components of $\mathbf{r}_i$, as shown in Equation (8); $\alpha_i$ is the angle of rotation and is also the modulus of $\mathbf{r}_i$.

$$\mathbf{r}_i = \begin{pmatrix} r_x & r_y & r_z \end{pmatrix}^T \tag{8}$$

When the ArUco numbered $i$ is detected, the transformation matrix $\mathbf{T}^{\mathbf{camera}}_{\mathbf{gimble}i}$ of the camera frame to the gimbal frame is shown in Equation (9).

$$\mathbf{T}^{\mathbf{camera}}_{\mathbf{gimble}i} = \begin{pmatrix} 0 & -1 & 0 & 0 \\ -1 & 0 & 0 & 0 \\ 0 & 0 & -1 & 0 \\ 0 & 0 & 0 & 1 \end{pmatrix} \begin{pmatrix} \mathbf{R}_{\mathbf{gimble}i} & 0 \\ 0 & 1 \end{pmatrix} \tag{9}$$

where $\mathbf{R}_{\mathbf{gimble}}$ is the rotation matrix corresponding to the Euler angle of the gimbal, as shown in Equation (10)

$$\mathbf{R}_{\mathbf{gimble}i} = \Re \begin{pmatrix} \varphi_{gi} \\ \theta_{gi} \\ \phi_{gi} \end{pmatrix} \tag{10}$$

The transformation matrix between the Lidar and the gimbal obtained by calibration is shown in Equation (11).

$$\mathbf{T}^{\mathbf{lidar}}_{\mathbf{gimble}} = \begin{pmatrix} 1 & 0 & 0 & x_g \\ 0 & 1 & 0 & y_g \\ 0 & 0 & 1 & z_g \\ 0 & 0 & 0 & 1 \end{pmatrix} \tag{11}$$

In this paper, a dynamic measurement noise covariance matrix of $\mathbf{T}^{\mathbf{lidar}}_{\mathbf{world}i}$ is proposed, which can be adjusted by observation of the ArUco numbered $i$ in the camera frame and image frame and shown as $\mathbf{R}_i(C_{i1}(\xi_i(\mathbf{r}_i, \mathbf{t}_i)), C_{i2}(d_i))$. Where, the rotation vector $\mathbf{r}_i$, the translation vector $\mathbf{t}_i$, and the global ArUco side length $l$ are used to calculate the influence factor $\xi_i(\mathbf{r}_i, \mathbf{t}_i)$ of the measurement noise covariance caused by the different measurement angles due to the pose. In addition, the distance $d_i$ from the center of gravity of the ArUco numbered $i$ to the optical axis is used as the influence factor of the measurement noise covariance caused by different positions of ArUco in the image frame. Before the operation of the navigation system and after the internal and external parameters of the camera are calibrated, the measurement noise covariance matrix needs to be calibrated. When calibrating, respectively change the above two influencing factors to measure the measurement noise, calculate the covariance, and then fit the functions $C_{i1}$ and $C_{i2}$ according to the results to complete the calibration.

In calculating the global ArUco factor, EKF is used to fuse IMU with the global pose obtained by global ArUco markers. The reason for choosing EKF is that under the conditions of use in this paper, compared with UKF and PF, EKF has similar result accuracy, occupies less computing resources, has low computational complexity, and has no strict initialization requirements. The UAV's position $\mathbf{p^w} = \begin{pmatrix} x & y & z \end{pmatrix}^T$ and attitude $\mathbf{\Phi^w} = \begin{pmatrix} \gamma & \theta & \varphi \end{pmatrix}^T$ in the world frame and the velocity $\mathbf{v^b} = \begin{pmatrix} v_x & v_y & v_z \end{pmatrix}^T$, angular velocity $\mathbf{\omega^b} = \begin{pmatrix} \varpi_x & \varpi_y & \varpi_z \end{pmatrix}^T$, and acceleration $\mathbf{a^b} = \begin{pmatrix} a_x & a_y & a_z \end{pmatrix}^T$ in the body frame (Front-Left-Upper) are taken as the system state $\mathbf{s} = \begin{pmatrix} \mathbf{p^w}^T & \mathbf{\Phi^w}^T & \mathbf{v^b}^T & \mathbf{\omega^b}^T & \mathbf{a^b}^T \end{pmatrix}^T$. The transformation matrix from the body frame to the world frame is denoted as $\mathbf{C}$, as shown in Equation (12), the relationship between the angular velocity $\mathbf{\omega^b}$ and the attitude $\mathbf{\Phi^w}$ is shown in Equation (13).

$$\mathbf{C} = \Re(\mathbf{\Phi^w}) \tag{12}$$

$$\mathbf{\omega^w} = \dot{\mathbf{\Phi}}^{\mathbf{w}} = \begin{pmatrix} \dot{\gamma} \\ \dot{\theta} \\ \dot{\varphi} \end{pmatrix} = \mathbf{R} \begin{pmatrix} \varpi_x \\ \varpi_y \\ \varpi_z \end{pmatrix} \tag{13}$$

where $\mathbf{R}$ is shown as:

$$\mathbf{R} = \begin{pmatrix} 1 & \sin\gamma \frac{\sin\theta}{\cos\theta} & \cos\gamma \frac{\sin\theta}{\cos\theta} \\ 0 & \cos\gamma & -\sin\gamma \\ 0 & \sin\gamma \frac{1}{\cos\theta} & \cos\gamma \frac{1}{\cos\theta} \end{pmatrix}$$

The prior estimation of the position, velocity, attitude, acceleration, and angular velocity at time k in the EKF are shown in Equations (14)–(18), where $\Delta t$ is the time interval between two adjacent states in the EKF. The state equation of the system is shown in Equation (19), where $\mathbf{A}$ is the state transition matrix, written from Equation (14) to

Equation (18), and $\mathbf{W_{k-1}}$ is the prior noise. The covariance estimation is shown in Equation (20), where $\mathbf{Q}$ is the prior error noise covariance matrix.

$$\widetilde{\mathbf{p}}_{\mathbf{k}}^{\mathbf{w}} = \mathbf{p}_{\mathbf{k-1}}^{\mathbf{w}} + \mathbf{C}\mathbf{v}_{\mathbf{k-1}}\Delta t + \mathbf{C}\mathbf{a}_{\mathbf{k-1}}\frac{\Delta t^2}{2} \tag{14}$$

$$\widetilde{\mathbf{v}}_{\mathbf{k}}^{\mathbf{b}} = \mathbf{v}_{\mathbf{k-1}}^{\mathbf{b}} + \mathbf{a}_{\mathbf{k-1}}\Delta t \tag{15}$$

$$\widetilde{\boldsymbol{\Phi}}_{\mathbf{k}} = \boldsymbol{\Phi}_{\mathbf{k-1}} + \mathbf{R}\boldsymbol{\omega}_{\mathbf{k-1}}\Delta t \tag{16}$$

$$\widetilde{\mathbf{a}}_{\mathbf{k}} = \mathbf{a}_{\mathbf{k-1}} \tag{17}$$

$$\widetilde{\omega}_{\mathbf{k}} = \omega_{\mathbf{k-1}} \tag{18}$$

$$\widetilde{\mathbf{s}}_{\mathbf{k}} = \mathbf{A}\mathbf{s}_{\mathbf{k-1}} + \mathbf{W}_{\mathbf{k-1}} \tag{19}$$

$$\widetilde{\mathbf{P}}_{\mathbf{k}} = \mathbf{A}\mathbf{p}_{\mathbf{k-1}}\mathbf{A}^{\mathbf{T}} + \mathbf{Q} \tag{20}$$

In the EKF calculation, first, calculate the measurement noise covariance of all ArUcos detected at time k, sort them, select the lowest $\mathbf{R}_i(C_{i1}(\xi_i(\mathbf{r}_i, \mathbf{t}_i)), C_{i2}(d_i))$, and record the corresponding ArUco number $i$ and global pose $\mathbf{T}_{\mathbf{world}i}^{\mathbf{lidar}}$ simultaneously. The global position and attitude observations are recorded as $\mathbf{p}_{\mathbf{zk}}$ and $\boldsymbol{\Phi}_{\mathbf{zak}}$, respectively, where $\mathbf{p}_{\mathbf{zk}}$ is the translation part of $\mathbf{T}_{\mathbf{world}i}^{\mathbf{lidar}}$, and the rotation matrix part of $\mathbf{T}_{\mathbf{world}i}^{\mathbf{lidar}}$ solves $\boldsymbol{\Phi}_{\mathbf{zak}}$. The IMU's acceleration, attitude, and angular velocity are recorded as $\mathbf{a}_{\mathbf{imu}}$, $\boldsymbol{\Phi}_{\mathbf{imu}}$, and $\omega_{\mathbf{imu}}$, respectively. The observation equation is shown in Equation (21), where $\mathbf{V}_{\mathbf{k}}(i)$ is the measurement noise, and its covariance is $\boldsymbol{\Sigma}_{\mathbf{k}}$. $\boldsymbol{\Sigma}_{\mathbf{k}}$ is obtained by $\mathbf{R}_i^{Euler}$ and $\mathbf{R}_{\mathbf{imu}}$, shown as Equation (22). Where $\mathbf{R}_i^{Euler}$ is $\mathbf{R}_i(C_{i1}(\xi_i(\mathbf{r}_i, \mathbf{t}_i)), C_{i2}(d_i))$ transformed into Euler angle form, and $\mathbf{R}_{\mathbf{imu}}$ is the measurement noise covariance of IMU, obtained through IMU calibration.

$$\mathbf{Z}_{\mathbf{k}} = \begin{pmatrix} \mathbf{p}_{\mathbf{zk}} \\ \boldsymbol{\Phi}_{\mathbf{zak}} \\ \mathbf{a}_{\mathbf{imu}} \\ \boldsymbol{\Phi}_{\mathbf{imu}} \\ \omega_{\mathbf{imu}} \end{pmatrix} = \mathbf{H}\mathbf{x}_{\mathbf{k}} + \mathbf{V}_{\mathbf{k}}(i) \tag{21}$$

$$\boldsymbol{\Sigma}_{\mathbf{k}} = \begin{pmatrix} \mathbf{R}_i^{Euler} & 0 \\ 0 & \mathbf{R}_{\mathbf{imu}} \end{pmatrix} \tag{22}$$

The correction equation for the state is shown in Equation (23), where $\mathbf{K}$ is the Kalman gain as shown in Equation (24). The correction equation for error covariance is shown in Equation (25).

$$\mathbf{s}_{\mathbf{k}} = \widetilde{\mathbf{s}}_{\mathbf{k}} + \mathbf{K}[\mathbf{Z}_{\mathbf{k}} - \mathbf{H}\widetilde{\mathbf{x}}_{\mathbf{k}}] \tag{23}$$

$$\mathbf{K} = \frac{\widetilde{\mathbf{P}}_{\mathbf{k}}\mathbf{H}^{\mathbf{T}}}{\mathbf{H}\widetilde{\mathbf{P}}_{\mathbf{k}}\mathbf{H}^{\mathbf{T}} + \sum_{\mathbf{k}}} \tag{24}$$

$$\mathbf{P}_{\mathbf{k}} = [\mathbf{I} - \mathbf{K}\mathbf{H}]\widetilde{\mathbf{P}}_{\mathbf{k}} \tag{25}$$

The global ArUco factor is taken as the initial value of the current state $\mathbf{X_k}$ in the factor graph. The global ArUco factor transformed from the position $\mathbf{p^w}$, attitude $\boldsymbol{\Phi}^{\mathbf{w}}$ part in the state $\mathbf{s_k}$ corrected by the EKF at time k and the corresponding parts of $\mathbf{p^w}$, $\boldsymbol{\Phi}^{\mathbf{w}}$ in the covariance matrix $\mathbf{P_k}$, as shown by (b) in Figure 3. In addition, when no ArUco has been detected within 3 s, the EKF process will be stopped and initialized, and will be restarted while ArUco are being detected again.

### 3.3. IMU Pre-Integration Factor

The process of IMU information processing and construction of the pre-integration factor used in this paper is the same as that in reference [24].

The measurement equations of the gyroscope and the accelerometer are shown in Equations (26) and (27), respectively.

$$\overline{\boldsymbol{\omega}}^{\mathbf{b}} = \boldsymbol{\omega}^{\mathbf{b}} + \mathbf{b}^{\mathbf{g}} + \mathbf{n}^{\mathbf{g}} \tag{26}$$

$$\overline{\mathbf{a}}^{\mathbf{b}} = \mathbf{R}_{\mathbf{bw}}(\mathbf{a}^{\mathbf{w}} + \mathbf{g}^{\mathbf{w}}) + \mathbf{b}^{\mathbf{a}} + \mathbf{n}^{\mathbf{a}} \tag{27}$$

where $\overline{\boldsymbol{\omega}}^{\mathbf{b}}$ and $\overline{\mathbf{a}}^{\mathbf{b}}$ are the measured values of the gyroscope and accelerometer, $\boldsymbol{\omega}^{\mathbf{b}}$ and $\mathbf{a}^{\mathbf{w}}$ are the truths of angular velocity and acceleration, $\mathbf{b}^{\mathbf{g}}$ and $\mathbf{n}^{\mathbf{g}}$ are the bias and noise of gyroscope, $\mathbf{b}^{\mathbf{a}}$ and $\mathbf{n}^{\mathbf{a}}$ are the bias and noise of accelerometer, and $\mathbf{R}_{\mathbf{bw}}$ is the transformation matrix from the world frame to body frame.

Assuming time $i \leq k < k + 1 \leq j$, the average values of angular velocity and acceleration in adjacent times $k$ and $k + 1$ are calculated using the median method, denoted as $\boldsymbol{\omega}$ and $\mathbf{a}$, respectively, as shown in Equations (28) and (30), respectively. The time interval between time $k$ and $k + 1$ is $\Delta t$. The attitude, position, speed, accelerometer deviation, and gyroscope deviation of the pre-integration from the time $i$ to time $k + 1$ are shown in Equations (29) and (31) to (34), respectively. When time $k + 1 = j$, it is the pre-integration from the time $i$ to time $j$.

$$\boldsymbol{\omega} = \frac{1}{2}\left[\left(\boldsymbol{\omega}^{\mathbf{b_k}} - \mathbf{b_k^g}\right) - \left(\boldsymbol{\omega}^{\mathbf{b_{k+1}}} - \mathbf{b_k^g}\right)\right] \tag{28}$$

$$\mathbf{q}_{\mathbf{b_i b_{k+i}}} = \mathbf{q}_{\mathbf{b_i b_k}} \otimes \begin{pmatrix} 1 \\ \frac{1}{2}\boldsymbol{\omega}\Delta t \end{pmatrix} \tag{29}$$

$$\mathbf{a} = \frac{1}{2}\left[\mathbf{R}_{\mathbf{b_i b_k}}\left(\mathbf{a}^{\mathbf{b_k}} - \mathbf{b_k^a}\right) + \mathbf{R}_{\mathbf{b_i b_{k+1}}}\left(\mathbf{a}^{\mathbf{b_{k+1}}} - \mathbf{b_k^a}\right)\right] \tag{30}$$

$$\mathbf{p}_{\mathbf{b_i b_{k+1}}} = \mathbf{p}_{\mathbf{b_i b_k}} + \mathbf{v}_{\mathbf{b_i b_k}}\Delta t + \frac{1}{2}\mathbf{a}\Delta t \tag{31}$$

$$\mathbf{v}_{\mathbf{b_i b_{k+1}}} = \mathbf{v}_{\mathbf{b_i b_k}} + \mathbf{a}\Delta t \tag{32}$$

$$\mathbf{b_{k+1}^a} = \mathbf{b_k^a} + \mathbf{n}_{\mathbf{b_k}}^{\mathbf{a}}\Delta t \tag{33}$$

$$\mathbf{b_{k+1}^g} = \mathbf{b_k^g} + \mathbf{n}_{\mathbf{b_k}}^{\mathbf{g}}\Delta t \tag{34}$$

The pre-integration factor is transformed from Equation (35) added to the factor graph as the measurement between two adjacent states, $\mathbf{X_i}$ and $\mathbf{X_j}$, as shown by (c) in Figure 3. The specific derivation process of error and covariance can refer to reference [11] and reference [23].

$$\begin{pmatrix} \mathbf{r_p} \\ \mathbf{r_v} \\ \mathbf{r_q} \\ \mathbf{r_{b^a}} \\ \mathbf{r_{b^g}} \end{pmatrix}_{15 \times 1} = \begin{pmatrix} \mathbf{R_{b_i w}}\left(\mathbf{p_{wb_j}} - \mathbf{p_{wb_i}} - \mathbf{v_i}\Delta\mathbf{t} + \frac{1}{2}\mathbf{g^w}\Delta t^2\right) - \mathbf{p_{b_i b_j}} \\ 2\left[\mathbf{q_{b_j b_i}} \otimes \left(\mathbf{q_{b_i w}} \otimes \mathbf{q_{wb_j}}\right)\right]_{xyz} \\ \mathbf{R_{b_i w}}\left(\mathbf{v_j^w} - \mathbf{v_i^w} + \mathbf{g^w}\Delta t\right) - \mathbf{v_{b_i b_j}} \\ \mathbf{b_j^a} - \mathbf{b_i^a} \\ \mathbf{b_j^g} - \mathbf{b_i^g} \end{pmatrix} \tag{35}$$

### 3.4. Lidar Factor

When collecting the laser point cloud, first perform motion compensation on each point, align the timestamp, and project a period of the point cloud onto a frame of the point cloud image, which is recorded as the frame $n$ [25]. Then, perform feature extraction on the frame point cloud image. Calculate the average distance $k_k$ from five points before and after a point $\mathbf{p_k}$ on each scan line to this point, as shown in Equation (36). $\|\mathbf{p_k}\|$ is the distance from the point $\mathbf{p_k}$ on the line to the center of the Lidar, and $\left\|\mathbf{p_k} - \mathbf{p_j}\right\|$ is the distance between the point $\mathbf{p_k}$ and the nearby points $\mathbf{p_j}$. If $k_k$ of point $\mathbf{p_k}$ is close to the average distance of points around $\mathbf{p_k}$, the curvature near point $\mathbf{p_k}$ is slight, and the terrain

probability is relatively smooth. The point $\mathbf{p_k}$ is generally on the plane and is recorded as $\mathbf{F_m}n$ as a planar feature. Conversely, if $k_k$ of point $\mathbf{p_k}$ differs significantly from the average distance of points around $\mathbf{p_k}$, the curvature near the point $\mathbf{p_k}$ is large, and the terrain probability changes abruptly. The point $\mathbf{p_k}$ is generally a corner point, recorded as $\mathbf{F_b}n$ as an edge feature [12].

$$k_k = \frac{1}{10 \cdot \|\mathbf{p_k}\|} \sum_{j \in [k-5, k+5], j \neq k} \left\| \mathbf{p_k} - \mathbf{p_j} \right\| \tag{36}$$

For the key frames selection, the first frame is used as the keyframe. For the rest, when determining whether frame $n$ is a keyframe, compare the covisibility relationship between the feature point set $\{\mathbf{F_m}n, \mathbf{F_b}n\}$ of frame $n$ and the feature points in the previous keyframe $\mathbf{k_m}$. If the change of the covisibility relationship is greater than the set threshold, set frame $n$ as a keyframe and record it as $\mathbf{k_{m+1}}$.

There are mainly the following steps when matching the features of two keyframes. Calculate the distance $\mathbf{d_{mk_{m+1}}}$ from the planar feature point $\mathbf{F_m}n$ in the latest keyframe $\mathbf{k_{m+1}}$ (the frame $n$) to the plane formed by the corresponding three adjacent planar feature points in the first five keyframes $\mathbf{k_m}$ to $\mathbf{k_{m-5}}$. Calculate the distance $\mathbf{d_{bk_{m+1}}}$ from the edge feature point $\mathbf{F_b}n$ in the latest keyframe $\mathbf{k_{m+1}}$ to the straight line formed by the corresponding adjacent edge feature points in the first five keyframes $\mathbf{k_m}$ to $\mathbf{k_{m-5}}$. Motion estimation is performed using $\left( \mathbf{d_{mk_{m+1}}} \quad \mathbf{d_{bk_{m+1}}} \right)^{\mathbf{T}}$ as a cost function to optimize the rotation translational changes in two keyframes. If there is no degradation [26] in the optimization solution, add state $\mathbf{X_{m+1}}$ to the factor graph, project $\mathbf{k_{m+1}}$ into the map, and take the optimization result as the measurement between state $\mathbf{X_m}$ and $\mathbf{X_{m+1}}$ in the factor graph. If the solution optimization process degenerates and the global ArUco is continuously detected, the keyframe $\mathbf{k_{m+1}}$ is projected into the map according to the pose solved by the global ArUco, but the optimization result is not added as the measurement between state $\mathbf{X_m}$ and $\mathbf{X_{m+1}}$ in the factor graph, as shown by (d) in Figure 3.

In addition, if the system reads the frame $n$ of Lidar and detects an ArUco numbered $i$ at the same time, even if the covisibility relationship is not lower than the threshold, this frame will be added as a keyframe $\mathbf{k_a}i$. In the subsequent continuous detection of the ArUco numbered $i$, if the covariance of the global ArUco factor is less than the covariance corresponding to the time when $\mathbf{k_a}i$ is detected, the keyframe $\mathbf{k_a}i$ will be updated. After the continuous detection of the ArUco numbered $i$ is completed, firstly, find the keyframe $\mathbf{k_m}$, which is generated by the covisibility relationship with the closest time of the keyframe $\mathbf{k_a}i$. Then, match $\mathbf{k_a}i$ with $\mathbf{k_m}$ and the adjacent keyframe $\mathbf{k_{m-2}}$ to $\mathbf{k_{m+2}}$ before and after $\mathbf{k_m}$. Finally, add $\mathbf{k_a}i$ to the map. In this paper, the discontinuity threshold for an ArUco detection is set to 3 s.

### 3.5. Global ArUco Loop Closure Factor

The global ArUco loop closure is used to correct the global pose. It adds the matching results with historical keyframes to the factor graph when the navigation system repeatedly runs to a similar position.

When the navigation system recognizes the continuous detection signal of the ArUco numbered $i$ for the $l$th time, it can be determined as a loop closure. The keyframes $\mathbf{k_{ai,1}}, \mathbf{k_{ai,2}}$ to $\mathbf{k_{ai,l-1}}$ and the collection of keyframes $\{\mathbf{k_{q-2}}, \mathbf{k_{q-1}}, \mathbf{k_q}, \mathbf{k_{q+1}}, \mathbf{k_{q+2}}\}$, $\{\mathbf{k_{w-2}}, \mathbf{k_{w-1}}, \mathbf{k_w}, \mathbf{k_{w+1}}, \mathbf{k_{w+2}}\}$ to $\{\mathbf{k_{e-2}}, \mathbf{k_{e-1}}, \mathbf{k_e}, \mathbf{k_{e+1}}, \mathbf{k_{e+2}}\}$ are marched with $\mathbf{k_r}$, and the optimization is used for motion estimation, respectively, where $\mathbf{k_{ai,l}}$ means the keyframe added by the loop closure detected of the ArUco numbered $i$ for the $l$th time. $\mathbf{k_q}, \mathbf{k_w}$ to $\mathbf{k_e}$, and $\mathbf{k_r}$ are the keyframes obtained from the covisibility relationship, with times that are closest to the keyframes $\mathbf{k_{ai,1}}, \mathbf{k_{ai,2}}$ to $\mathbf{k_{ai,l-1}}$, and $\mathbf{k_{ai,l}}$, respectively. If the optimizations have solutions and do not degenerate, the optimization solutions will be added to the factor graph as loop closure factors and as measurements between state $\mathbf{X_r}$ and states $\mathbf{X_q}, \mathbf{X_w}$ to $\mathbf{X_e}$, as shown by (e) in Figure 3.

## 4. Experiment

The navigation system test uses Velodyne-16 surround Lidar, MTI-300 inertial measurement unit, Intel NUC8 onboard computer, ZENMUSE X5S gimbal camera, and DJI Matrice200 RTK unmanned flight platform. The hardware and the test platform of the navigation system are shown in Figure 4. The Velodyne-16 surround Lidar has a vertical resolution of 2°, a horizontal resolution of 0.2°, a field angle of view of 30° vertical and 360° horizontal, a maximum detection distance of 100 m, a detection accuracy of 3 cm, and a set speed of 600 rpm during the test. The gyroscope of Mti-300 has a maximum range of 450°/s, an initial deviation of 0.2°/s, and an operation deviation of 10°/h. The accelerometer of Mti-300 has a maximum range of 200 m/s$^2$, an initial deviation of 0.05 m/s$^2$, and an operation deviation of 15 μg. The camera of ZENMUSE X5S uses a 4/3″ CMOS sensor, with a field angle of view of 72°, maximum image resolution of 5280 × 3956, and real-time image resolution of 1280 × 960. The Gimbal of ZENMUSE X5S has an angle jitter of ±0.01°. The dictionary DICT_6 × 6_50 is selected for global ArUcos. The length of the side of the ArUco markers is 0.8 m. The area size of the flight test environment is 200 m × 43 m × 18 m, and the flight speed is set around 1 m/s. The algorithm of the navigation system is implemented in C++ and runs on an Ubuntu 18.04 operating system based on ROS architecture.

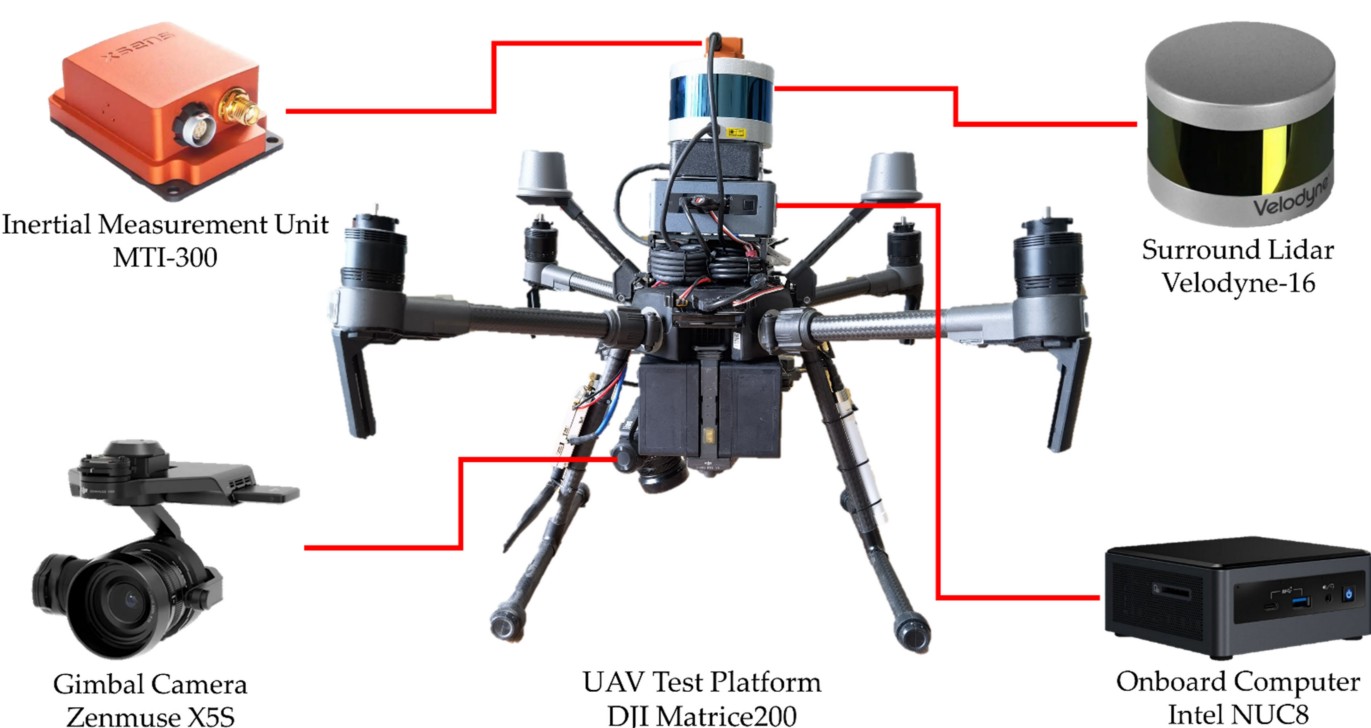

**Figure 4.** The hardware and the UAV platform.

After the navigation system is installed on a new carrying platform each time, the IMU shall be calibrated under the operation condition of the whole carrying system. After the MTI-300 is installed on the UAV platform, turn on all equipment and keep it still for 2 h. After the calibration, the average bias of the accelerometer and gyroscope measured is added to the factor graph as the initial value, and the measured noise covariance is added to EKF.

This chapter mainly introduces the calibration experiment before the fitting of the global ArUco dynamic measurement noise covariance matrix, the tests of the navigation system in the working condition, and the experiment on the navigation system accuracy.

### 4.1. The Calibration of Global ArUco Dynamic Measurement Noise Covariance

The global ArUco measurement noise includes the noise caused by different positions of ArUco in the image frame. Therefore, before calibration of the measurement noise covariance of global ArUco, it is necessary to calibrate the internal and external parameters of the camera.

During the calibration process, the working conditions are scaled proportionally. The 5 m × 5 m × 1.8 m Vicon system is used to measure the pose of the ArUco, the platform, and the sensors on it, and the measurement noise of the global ArUco system can be calibrated. In the test, Vicon markers are pasted on the Lidar, camera, and the flat plate where the ArUco marker is located so that the system can measure the real pose of the frames corresponding to the above devices. The schematic layout of the test equipment and the pasting position of Vicon markers are shown in Figure 5. Under working conditions, the average vertical distance between the camera and ArUco is about 16 m. Limited to the experimental conditions, the distance between the ArUco and camera imaging plane is 1 m during calibration to simulate the situation near the working condition of 16 m, and the ArUco side length is proportionally scaled to 0.05 m. The pixel plane is divided into 15 sampling areas. The schematic diagram of the projection of the sampling area in the pixel frame and image frame is shown in Figure 6. Due to symmetry, only 6 sampling areas in the first quadrant of the image frame are selected for calibration. Three conditions of 0.75 m, 1 m, and 1.25 m are selected as the distance from ArUco to the camera, and 25 combinations of roll angle and pitch angle $\pm 40°$, $\pm 20°$, and $0°$ are selected as the conditions of the ArUco attitude. In the calibration experiment, the global ArUco measurement noise is calibrated using a combination of the above different conditions in each sampling area. The ArUco pose in each sampling area is shown in Figure 7, and the detection of ARCUO in two experiments is shown in Figure 8. There are 450 states in the calibration experiment. After the experiment, the covariance matrix $\mathbf{R}_i(C_{i1}(\xi_i(\mathbf{r}_i, \mathbf{t}_i)), C_{i2}(d_i))$ is fitted according to the results of each state $\mathbf{r}_i$, $\mathbf{t}_i$, $d_i$, and measurement noise. In each sampling area of the first quadrant, when the pitch angle and roll angle of ArUco are 0 degrees and the distance from the camera is 0.75 m, the covariance calculated by calibration is shown in Table 1. It can be observed that the noise of different pose components obtained with different sampling positions has a large difference between the value and the changing trend. If the same covariance is used to replace all samples, the accuracy of the whole navigation system will be reduced.

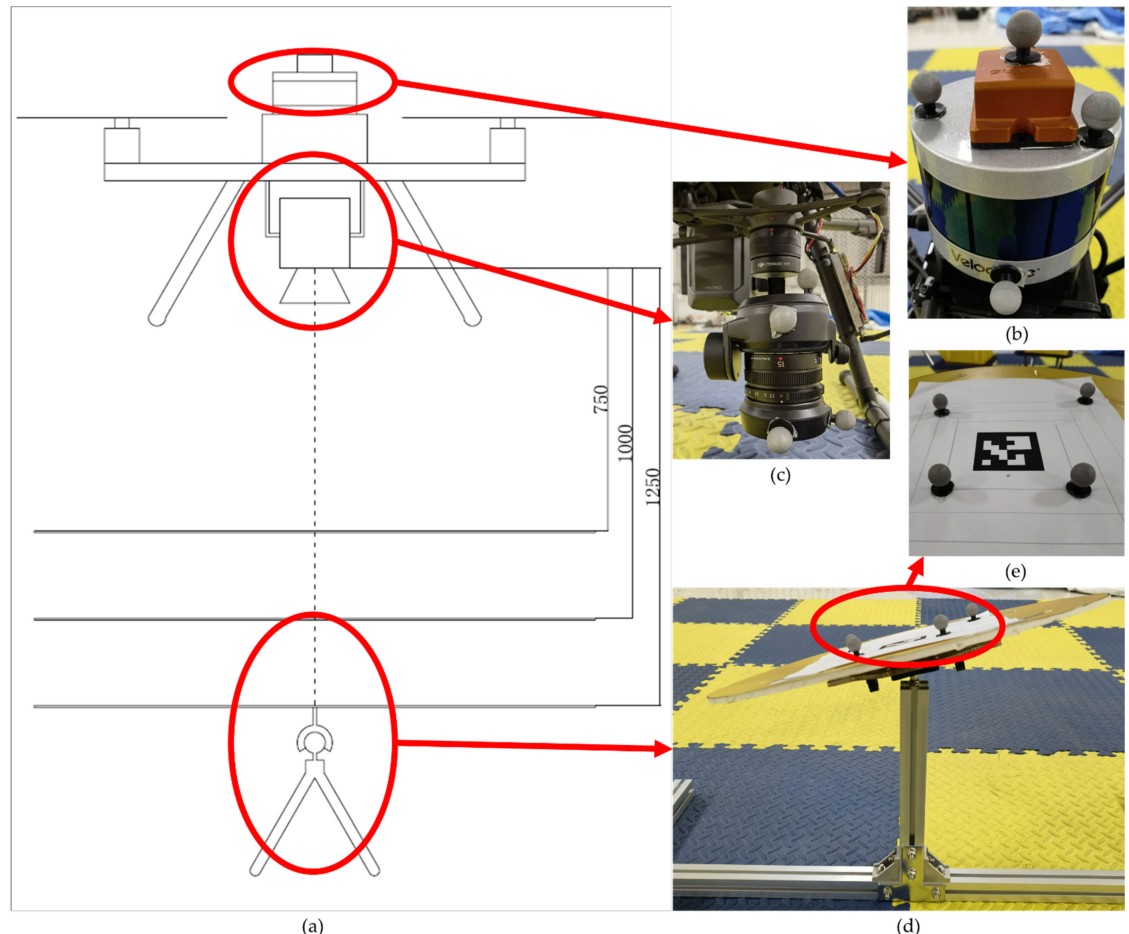

**Figure 5.** The schematic layout of test equipment and the pasting position of Vicon markers. (**a**) shows the test equipment, including the Lidar-IMU part, camera part on the UAV, and the installation of the ArUco marker on the ground. (**b**,**c**,**e**) are the pasting position of the Vicon mark near the Lidar-IMU part, the camera part, and the ArUco marker, respectively. (**d**) is the ball joint support of the ArUco marker.

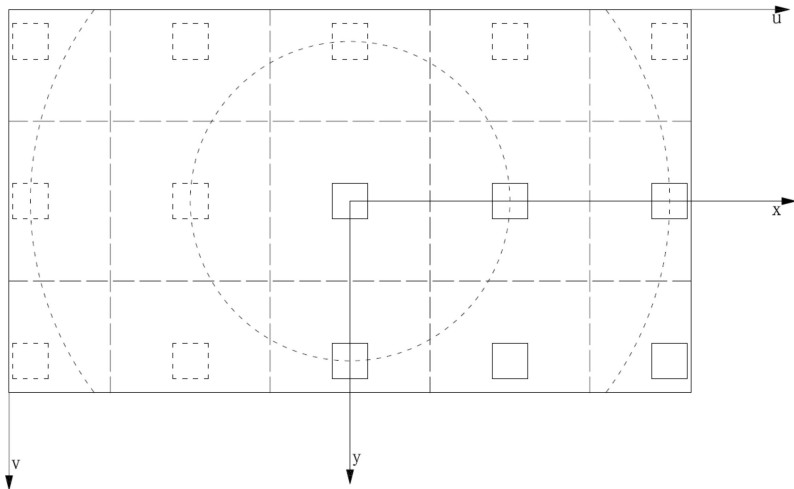

**Figure 6.** The projection of the sampling area in the pixel frame and image frame. The *x*-axis, *y*-axis constitute the image frame, and the u-axis and v-axis constitute the pixel frame.

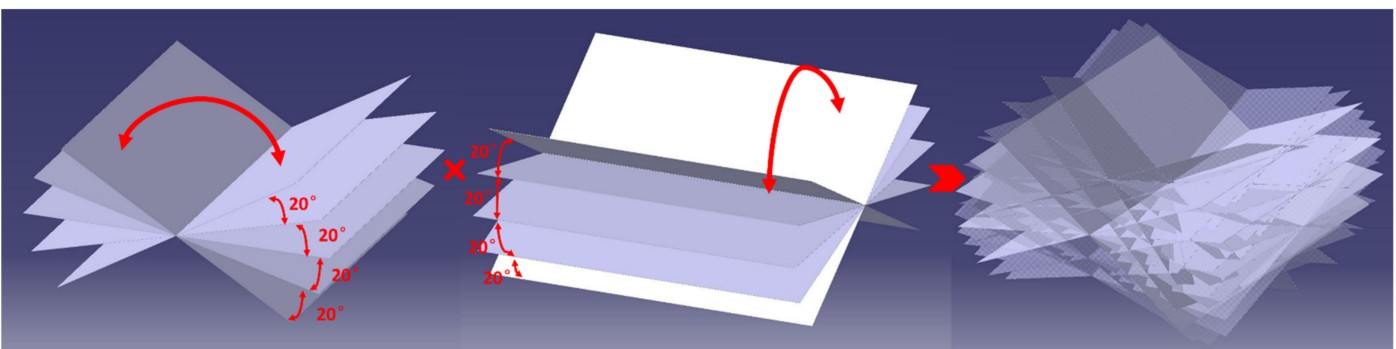

**Figure 7.** The schematic of ArUco pose in each sampling area.

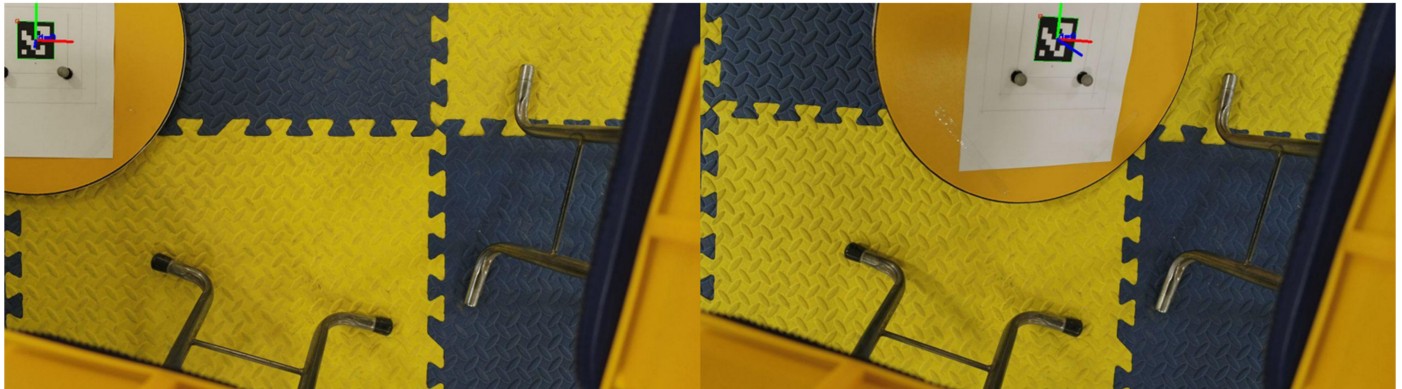

**Figure 8.** The detection of ARCUO in two experiments.

**Table 1.** The covariance in each sampling area of the first quadrant when roll and pitch angle of ArUco are both 0 degrees. X, Y represents the number of *x*-axis, and *y*-axis sampling areas, respectively.

| y \ x | 1 | 2 | 3 |
|---|---|---|---|
| 1 | $4.8 \times 10^{-8}$ | $1.8 \times 10^{-7}$ | $1.7 \times 10^{-6}$ |
|  | $2.3 \times 10^{-7}$ | $4.9 \times 10^{-8}$ | $5.9 \times 10^{-8}$ |
|  | $3.9 \times 10^{-5}$ | $5.7 \times 10^{-6}$ | $1.3 \times 10^{-5}$ |
|  | $5.5 \times 10^{-3}$ | $6.4 \times 10^{-4}$ | $1.6 \times 10^{-3}$ |
|  | $7.7 \times 10^{-3}$ | $2.5 \times 10^{-3}$ | $2.7 \times 10^{-3}$ |
|  | $5.2 \times 10^{-5}$ | $3.3 \times 10^{-5}$ | $5.8 \times 10^{-5}$ |

**Table 1.** *Cont.*

| x \ y | 1 | 2 | 3 |
|---|---|---|---|
| 2 | $7.8 \times 10^{-8}$ | $7.4 \times 10^{-7}$ | $5.2 \times 10^{-6}$ |
| | $1.0 \times 10^{-6}$ | $9.3 \times 10^{-7}$ | $1.6 \times 10^{-6}$ |
| | $2.9 \times 10^{-5}$ | $2.4 \times 10^{-5}$ | $4.6 \times 10^{-5}$ |
| | $1.8 \times 10^{-3}$ | $8.8 \times 10^{-4}$ | $2.6 \times 10^{-4}$ |
| | $2.5 \times 10^{-2}$ | $2.6 \times 10^{-2}$ | $2.6 \times 10^{-2}$ |
| | $9.3 \times 10^{-4}$ | $1.1 \times 10^{-3}$ | $1.8 \times 10^{-3}$ |

*4.2. The Tests of the Navigation System in the Working Condition*

The navigation system introduced in this paper is used to navigate UAVs in the GNSS-denied environment of known large scenes. Therefore, this paper selects the actual working conditions that meet the above conditions to test the system described in this paper and compare it with common navigation algorithms. The test site is the dry coal shed of the first phase of Fengcheng Thermal Power Plant in Fengcheng City, Jiangxi Province. The site environment and test flight photo are shown in Figure 9. This working condition requires the UAV to complete the indoor zigzag reciprocating survey and mapping line in the dry coal shed and accurately know the absolute position of the UAV in the dry coal shed scene, so as to facilitate the flight and subsequent visual measurement. In this paper, half of the dry coal shed is used as the test area.

In this test, the X-direction refers to the north, the Y-direction refers to the west, and the Z-direction refers to the sky. The origin is located at the plane of the coal storage site of the coal shed.

This paper records the original point cloud scanned by Lidar, the image taken by the camera, and the data measured by other sensors during two uninterrupted operations of UAV under working conditions. Use LOAM [12], LIO-SAM [16], and ArUco_LIO, the navigation system described in this paper, to solve the above types of information and compare them. The test flight used the manual flight mode, set the cruise altitude of 16.5 m, and had the flight path range of 90 m × 30 m. The trajectory of two uninterrupted operations calculated by ArUco_LIO is shown in Figure 10. The horizontal projection of the trajectory calculated by each navigation system is shown in Figure 11. The frame in Figure 11 is the world frame drawn according to the known environment. In Figure 11a, "range" represents the ground range of the dry coal shed in the real scene. "ArUco" means the global ArUco markers with different numbers in the scene in advance. Except for the take-off point and the two nearby ArUcos, all of them are arranged on the sidewalk 1.5 m high from the take-off point. The y-axis coordinate is −14.8 m, the distance between adjacent markers is 10 m, and no marker is placed at x = 70 m due to fixed equipment. "LOAM," "LIO-SAM," and "ArUco_LIO", respectively, represent the horizontal projection of the trajectory calculated by the three methods. "rotate" represents the result of manually rotating the solution of ArUco_LIO to a state similar to the other two algorithms. In Figure 11b, "LOAM," "LIO-SAM," and "ArUco_LIO", respectively, represent the projection

of the three algorithm trajectories in the *x-z* plane. The *z*-axis variation curve of each algorithm with time is shown in Figure 12.

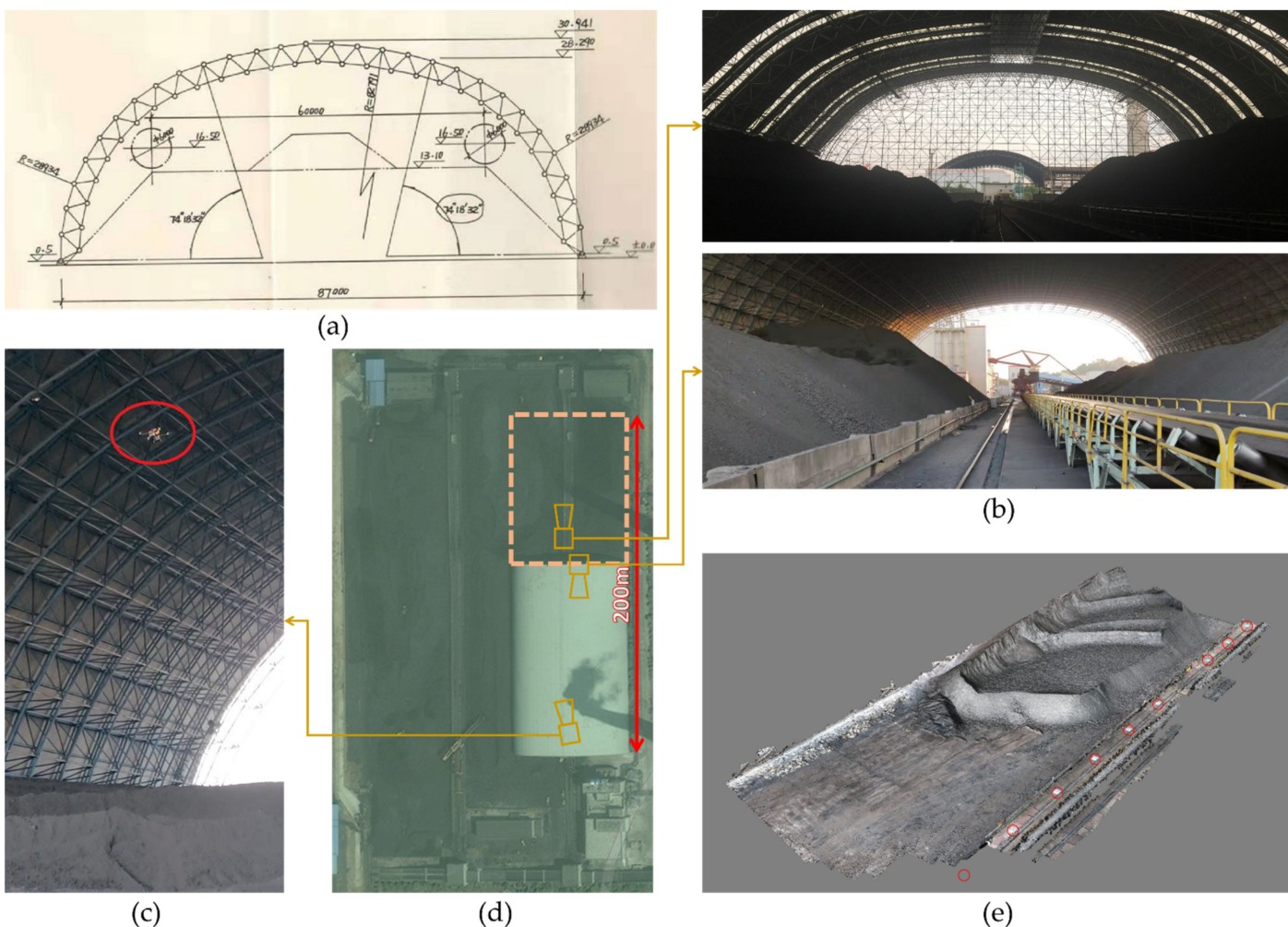

**Figure 9.** The test environment and test flight photo. (**a**) is the schedule drawing of dry coal shed roof. (**b**) is the site environment photo. (**c**) is the photo during the test flight. (**d**) is the satellite image of the dry coal shed. Although the square part is in the open air in the satellite image, the roof was built during the flight test. (**e**) is the three-dimensional map of the flight test site obtained by the visual three-dimensional mapping method irrelevant to this paper. The part circled in red is the global ArUco. Four markers near the takeoff point and on the sidewalk are not shown in the figure due to the scope of the drawing.

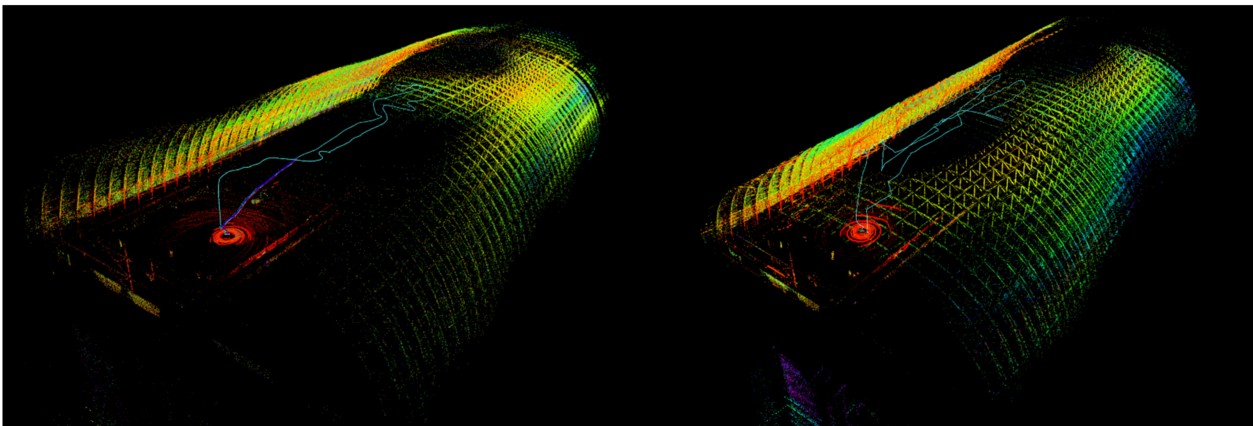

**Figure 10.** The trajectory of two uninterrupted operations calculated by ArUco_LIO.

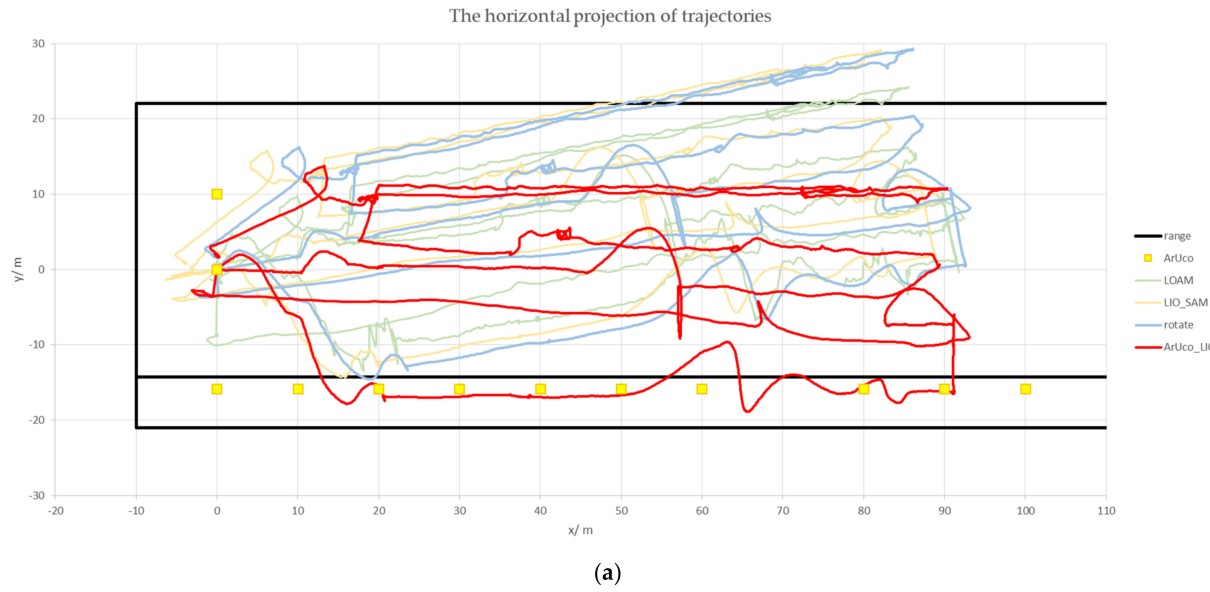

(**a**)

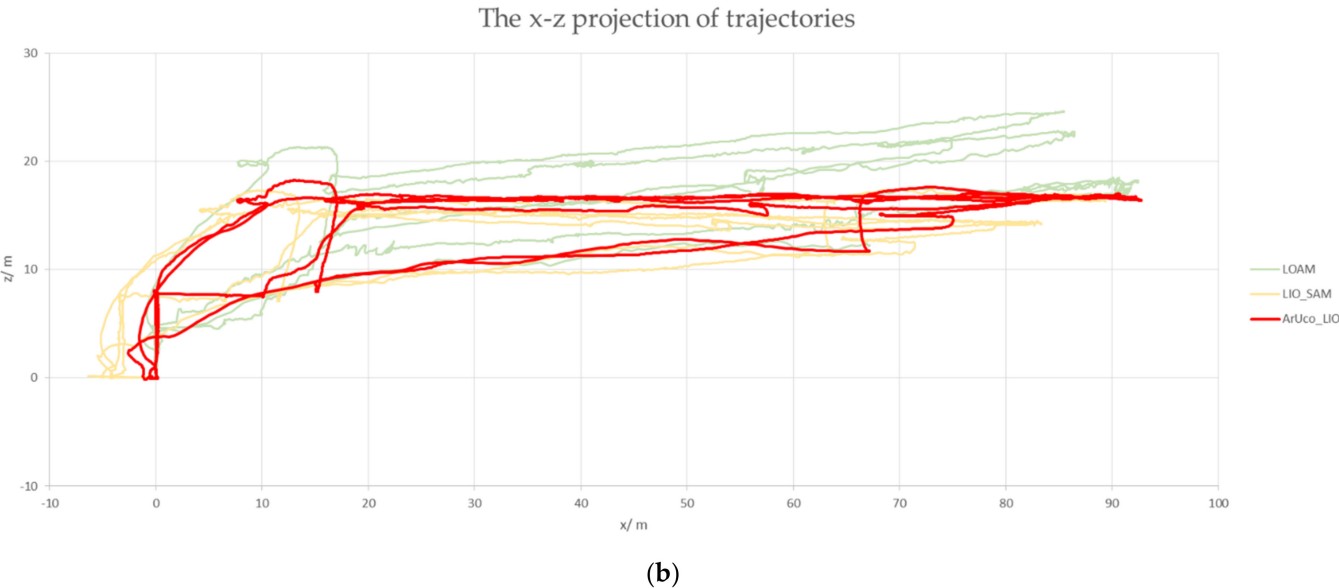

(**b**)

**Figure 11.** The trajectory calculated by each navigation system. (**a**) is the horizontal projection of the trajectory; (**b**) is the x-z projection of the trajectory.

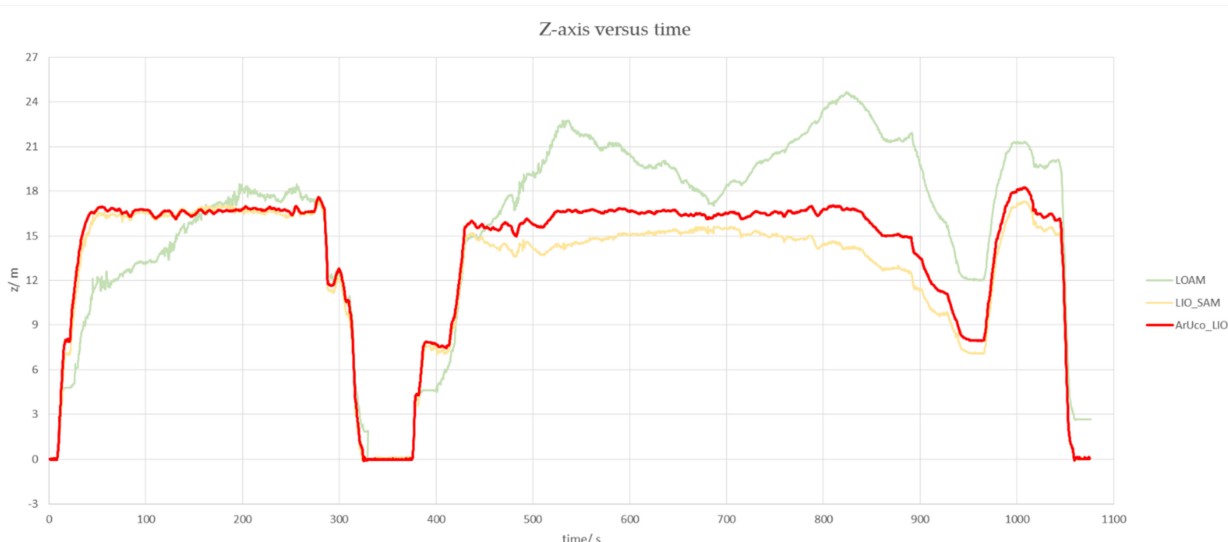

**Figure 12.** The z-axis variation curve with time.

As can be observed from Figure 11a, the trajectory obtained by the LOAM and LIO_SAM algorithms is affected by the UAV initial pose and the deviation of the installation angle of the Lidar. Because the algorithm does not have the function of correcting the trajectory rotation deviation, the calculated trajectory is far from the actual flight trajectory. Those navigation systems greatly increase the risk of UAV collision in the application, cannot meet the subsequent tasks such as visual measurement, and do not meet the needs of working conditions. By comparing the trajectory of "rotate" with other algorithms, it is found that LOAM degenerates soon after takeoff; it can be observed from the routes near the takeoff point and x = 90 m that LIO_SAM drifted near the takeoff point and was not repaired in the subsequent navigation. In contrast, the ArUco_LIO described in this paper completes the global optimization of navigation through global ArUco, which significantly corrects the direction of the route and corrects the accumulated error.

It can be observed from Figure 11b that after the degradation of the LOAM, it has a considerable drift in the Z direction. LIO_SAM algorithm also impacts the direction of the frame after drifting near takeoff, resulting in the overall route bow down and inaccurate altitude. The ArUco_LIO described in this paper is corrected through the global ArUco, and the navigation system maintains a relatively stable and accurate output. It can be more clearly observed from Figure 12 that when the UAV is cruising at 16.5 m altitude, the output of the navigation system is stable without a large offset. Moreover, its response speed is higher than the other two navigation algorithms.

### 4.3. The Experiment on the Navigation System Accuracy

Since the navigation system is suitable for a large-scale denial environment, there is no way to set up such a large Vicon system to provide a true value. Therefore, this paper chooses the outdoor GNSS environment and uses the results of GNSS-RTK+VIO of the DJI flight platform as the true value to test the accuracy of the navigation system. The GNSS-RTK+VIO positioning accuracy given by DJI is horizontal 1 cm + 1 ppm (RMS) and vertical 2 cm + 1 ppm (RMS). (1 ppm: for every 1 km increase, the accuracy will become 1 mm worse.) Due to the outdoor influence on the Lidar field of view, the UAV platform carried the navigation system flow several times at a height of only about 3 m and a range of $20 \times 5 \text{ m}^2$ during the test. We also reduced the size of the ArUco markers to a 0.16 m side length with the decrease in flight altitude. The satellite map of the test site and the approximate range of the flight are shown in Figure 13a. The photos of the original environment of the site are shown in Figure 13b. The trajectory of a flight is shown in Figure 13c.

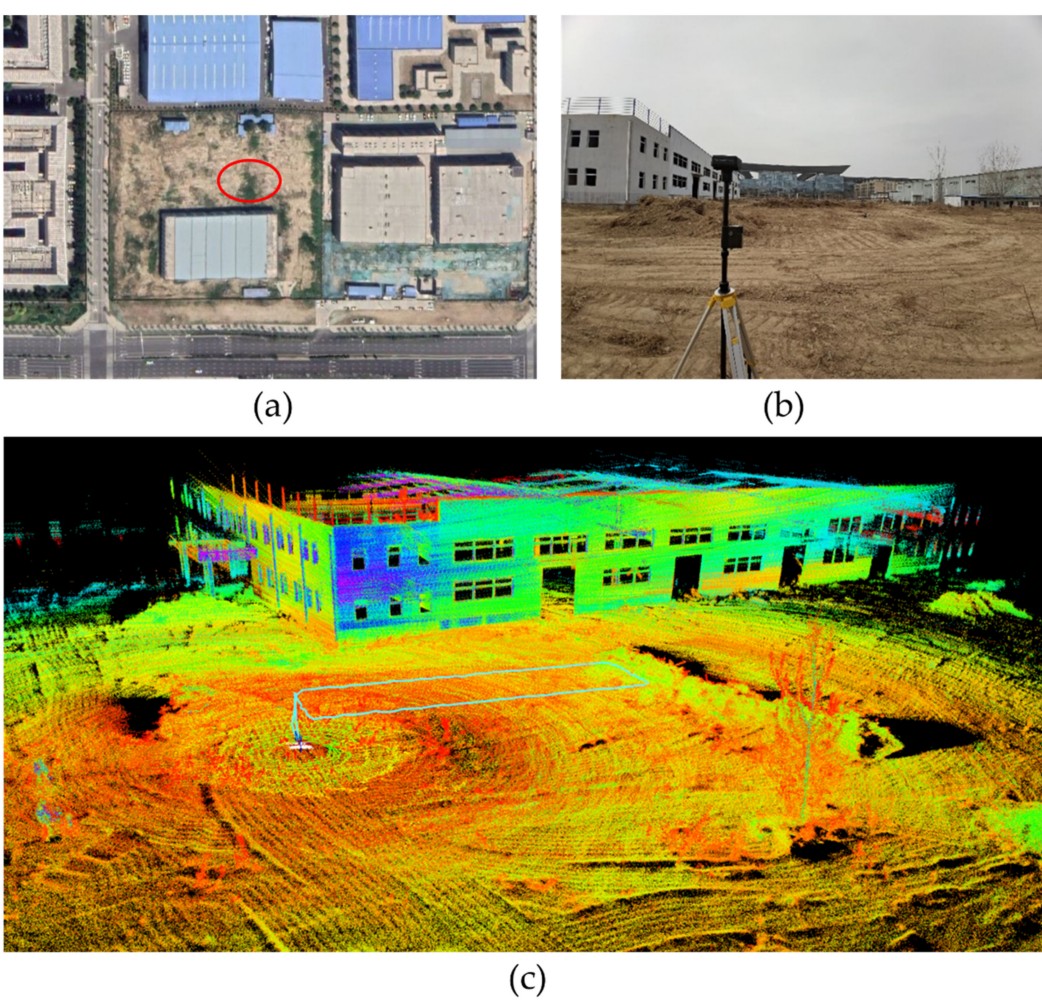

**Figure 13.** The photos of the test area. Where (**a**) is satellite map of the test site, the red circle is the approximate test area. (**b**) is the original environment of the test site. (**c**) is the flight path of a test in the point cloud map.

This paper converts several flights' GNSS-RTK+VIO longitude and latitude data into a metric system. The conversion is from west to x, south to y, and up to Z in this test. The converted results are arranged in the time sequence of the test and compared with ArUco_LIO. The horizontal projection of the track is shown in Figure 14, and the changes in X, y, and Z directions of the path with time are shown in Figure 15a–c, where red is the trajectory of ArUco_LIO, and light green is the true value. It can be observed from the image that the trajectory of ArUco_LIO is in good agreement with the true trajectory.

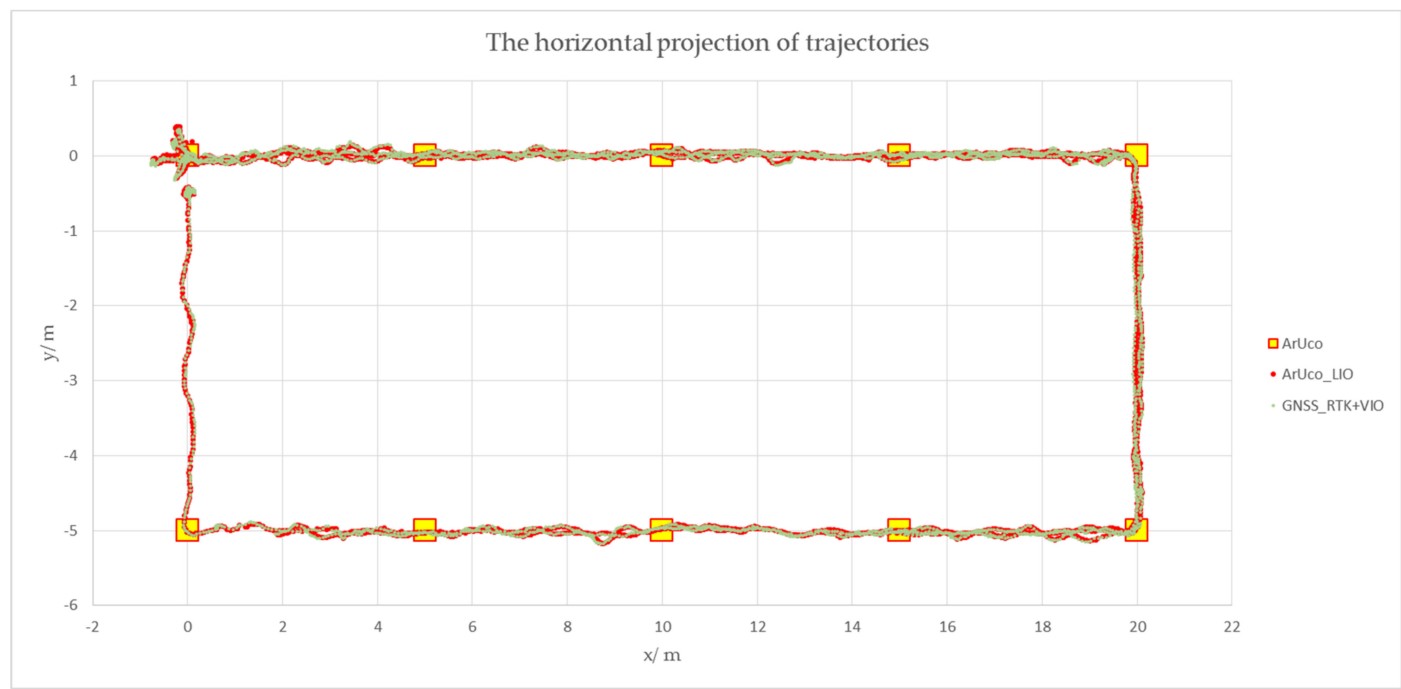

**Figure 14.** The horizontal projection of the track.

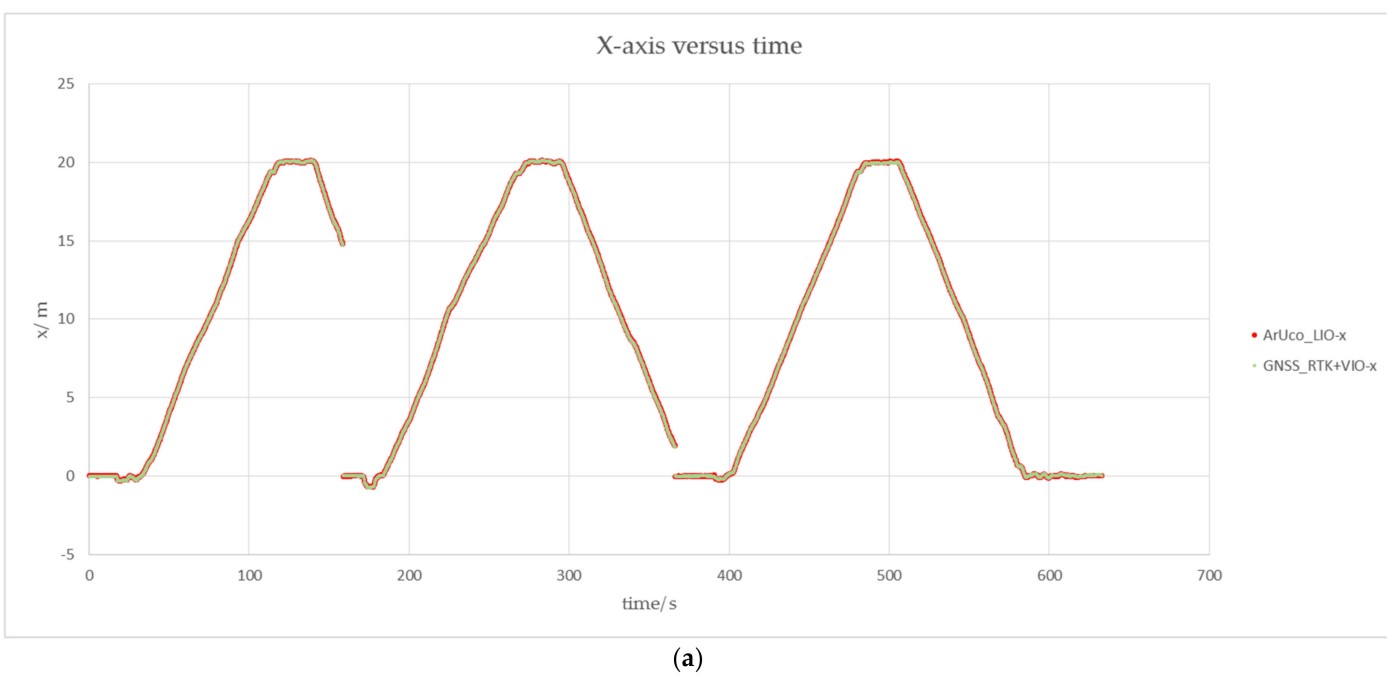

(**a**)

**Figure 15.** *Cont.*

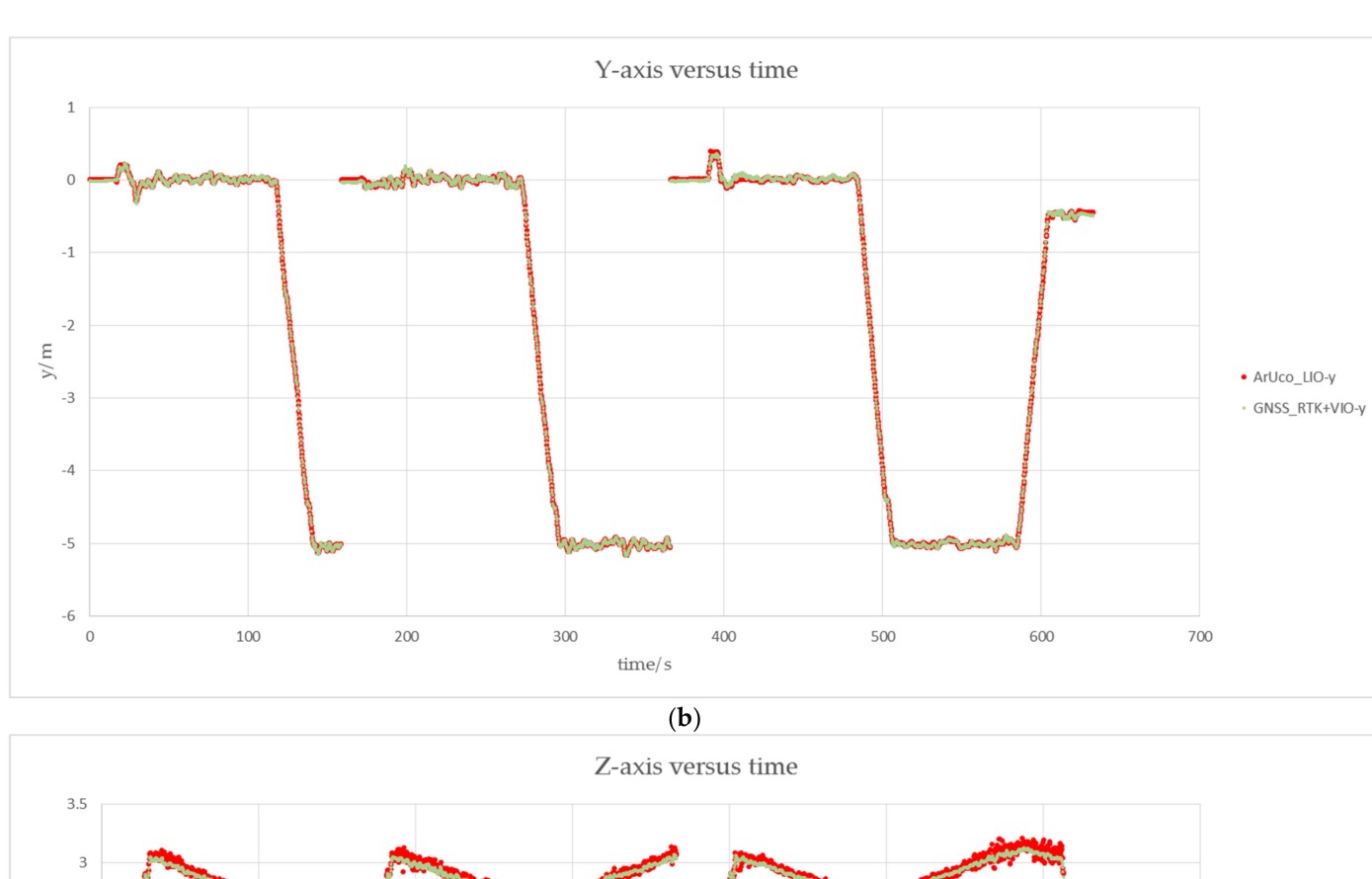

(**b**)

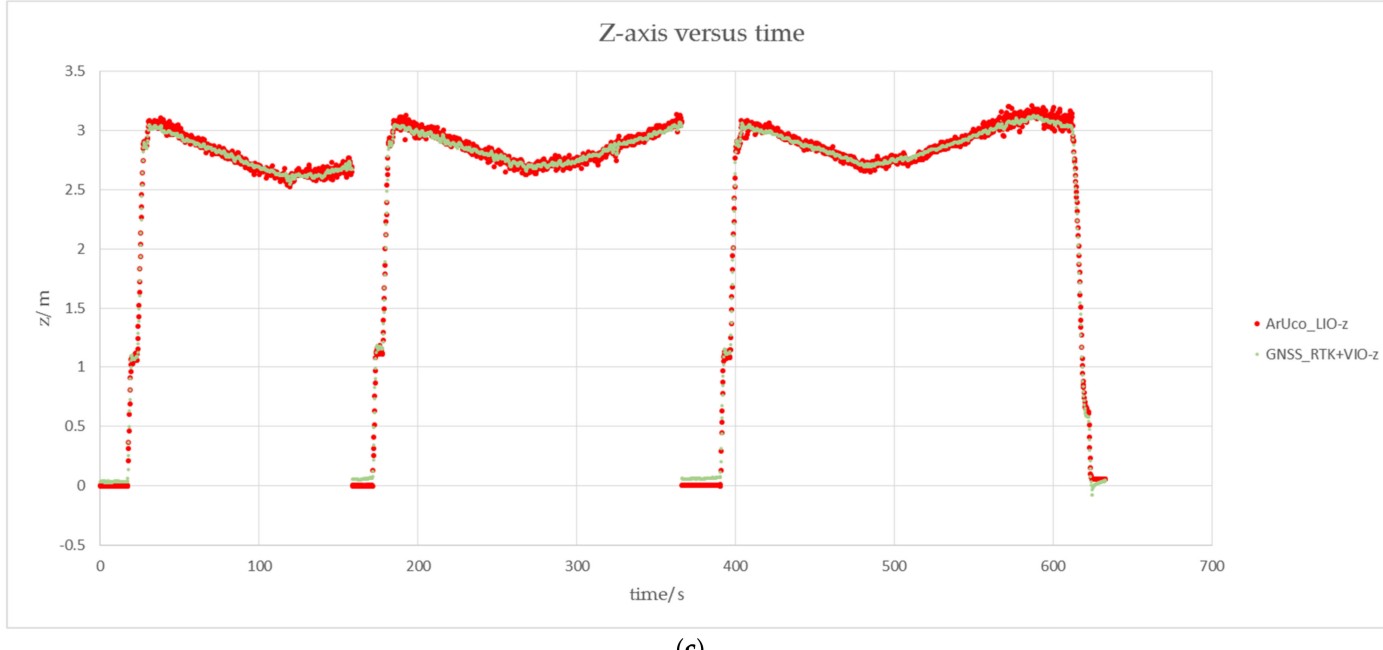

(**c**)

**Figure 15.** The changes in X, y, and Z directions of the path with time. Where (**a**) is the x-axis variation curve with time. (**b**) is the y-axis variation curve with time. (**c**) is the z-axis variation curve with time.

The scatter diagram of the navigation system positioning error is shown in Figure 16. The red scatters "error" and "error-XY" are the distance error between the ArUco_LIO and the true value of the three-dimensional position and the horizontal projection position, with DRMS of 0.04802 m and 0.03407 m, respectively. The blue scatters, "error-x," "error-y," and "error-z" are, respectively, the absolute errors in the X, y, and Z directions of ArUco_LIO in the test, with an RMS of 0.02876 m, 0.01826 m, and 0.03385 m. Therefore, under this working condition, the horizontal accuracy of ArUco_LIO is 0.03407 m, and the vertical accuracy is 0.03385 m.

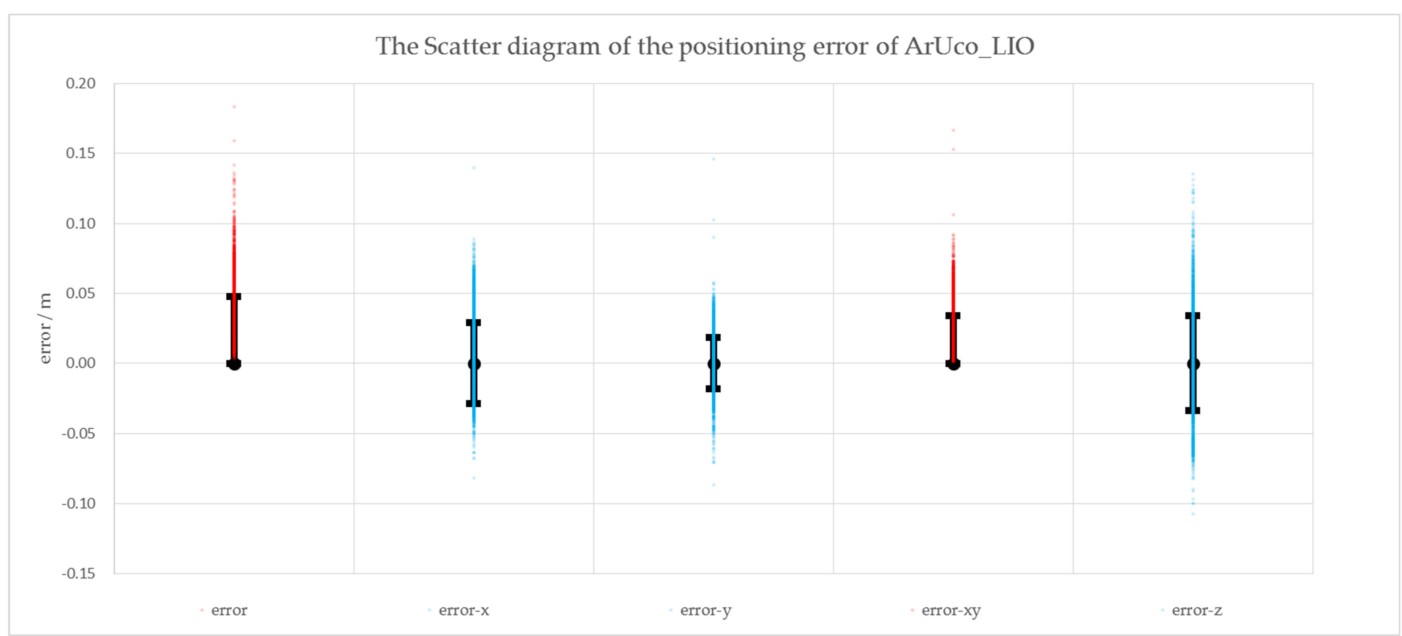

**Figure 16.** The Scatter diagram of the positioning error of ArUco_LIO.

## 5. Conclusions

This paper designs a Lidar navigation system based on global ArUco. The system can be widely used for high-precision navigation of UAVs in large indoor places with known GNSS-denied environments, such as factories, workshops, and dry coal sheds. Adding the global ArUco factor, whose confidence varies with sampling in the back-end, solves the problem of accurate positioning in the GNSS-denied environment, as well as efficiently solves the stability of the loopback detection. Furthermore, it also ensures the sustainable and stable operation of the system in the case of failure of the Lidar motion solution.

In this paper, the UAV platform is used to collect the data set under the working condition of the dry coal shed of the power plant, which is one of the suitable ranges of the system. The data set is used to thoroughly evaluate the system and the other two Lidar algorithms and compared with other algorithms in terms of a navigation effect according to the actual operation requirements. The results demonstrate that compared with LOAM and LIO-SAM, ArUco-LIO can work more accurately and stably in large-scale known GNSS-denied environments and can be used as a reliable navigation system for UAV in these scenes. This paper also tests the accuracy of the navigation system with GNSS-RTK+VIO as the true value in the GNSS environment and counts the errors. The results demonstrate that ArUco-LIO has high precision and can reliably provide navigation data for UAVs in the known GNSS-denied environment.

**Author Contributions:** Conceptualization, Z.Q. and R.J.; Data curation, Z.Q., J.L. and Z.Z.; Formal analysis, Z.Q.; Investigation, Z.Q. and R.J.; Methodology, Z.Q., R.J. and J.L.; Project administration, Z.Q., D.L. and R.J.; Resources, D.L. and R.J.; Software, Z.Q., J.L. and Z.Z.; Supervision, D.L. and R.J.; Validation, Z.Q., J.L. and Z.Z.; Visualization, Z.Q., J.L. and Z.Z.; Writing—original draft, Z.Q.; Writing—review & editing, Z.Q. and R.J. All authors have read and agreed to the published version of the manuscript.

**Funding:** This research received no external funding.

**Institutional Review Board Statement:** Not applicable.

**Informed Consent Statement:** Not applicable.

**Conflicts of Interest:** The authors declare no conflict of interest.

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
