# Peer review of "A Global ArUco-Based Lidar Navigation System for UAV Navigation in GNSS-Denied Environments"

_aerospace, doi:10.3390/aerospace9080456_

Round 1

Reviewer 1 Report

Thank you for the updated article, however, I would expect more detailed response to the comments after resubmission. There are no explanations or corrections for several comments. I have several further comments on the new section as well

Reviewer #1, 5) Why there are dots between γ, θ and φ an not between the omegas in 3-13?
There equation has not been changed, nor explanatio has been provided.

Reviewer #1, 6)
The dt has been changed to delta t, however, its meaning is still not specified.

Reviewer #1, 8) in 3-27 and 3-30 it looks like the quaternion is multiplying a bracket with 3-vectors. I believe that the rotation y quaternion should be applied. Maybe it is only necessary to refine or define the notation, however, the current state is misleading.

The multiplication still takes place, the quaternion rotation should be q*x*(q^-1), however. No explanation was provided.

- The comparison with the GNSS-RTK+VIO substantially improved the paper. I have still several comments/questions:

1) what is the expected or declared accuracy of the GNSS-RTK+VIO system?

2) why are several runs overlapped in fig 4-11?

3) why there are sudden jums and obvious restarts in the position vs time graphs (Fig 4-12)?

4) There is probably a flaw in evaluation of positioning error 4-13.
- evaluating mean and STD of the "error" and "error-xy" is misleading. They are always non-negative values, RMS error is appropriate measure comparabel to the STDs of the x,y and z errors.
Certainly, the error bars for "error" and "error-xy" are a misinterpretation of the results.
- The resulting errors seem comparable to the accuracy of RTK, could you elaborate on this?

In conclusion there is:"This paper also tests the accuracy of the navigation system with GNSS-RTK as the true value in the GNSS environment and counts the errors.I believe the wording is not correct, since the ArUco-based navigation is not used WITH GNSS-RTK, but is COMPARED TO GNSS-RTK+VIO system.

Author Response

Since only one word file can be submitted, I added coverletter at the end of the article.

Reviewer 2 Report

I am satisfied with revision.

Round 2

Reviewer 1 Report

Thank you for resolving or clarifying all the previous comments. The cover letter with expanations was useful, and led to better understanding of the changes.

This manuscript is a resubmission of an earlier submission. The following is a list of the peer review reports and author responses from that submission.

Round 1

Reviewer 1 Report

I believe the quality of the article is quite good, however, a few comments should be resolved:
1) The ArUco and its outputs use should be introduced more thoroughly in the introductory part of the article.

2) The ArUco is not referenced properly - its only and first reference is [22]. It should be referenced at its first occurence in the text - certainly within introduction. The ArUco developers suggest how to reference the library on the project website.

3) page 3, line 108-109: There is an object missing in the sentence. What is the thing that shoulb he high precision low-frequency?

4) Equations 3-3, 3-10 and 3-12 are very similar. I believe that it would be better to define an operator that constructs a rotation matrix from 3 angles and use this operator instead of always rewriting the rotation matrix in full.

5) Why there are dots between γ, θ and φ an not between the omegas?

6) What is the interpretation of dt in eqs 3-14 to 3-17. In continuous time, I would expect the integral over a time interval to be there. If it is the time between filter epochs, i would suggest using Δt, and more importantly, specifying its meaning. Maybe it is continuous and the integral marks are missing - I am not sure. Please resolve this.

7) The use of rotation matrices, quaternions and euler angles is somehow confusing:
Above 3-13 it is said thatrelationship between angular velocity and attitude quaternion will be described and there are no quaternions in the equation. Then the quaternions appear in 3-16. I believe the use of quternions and rotation matrices should be calrified and made somehow consistent throughout the article.
Moreover, qw contains 3 angles on line 237, whilst the quaternions in Sec 3.3 are aare denoted by q as well. This is very confusing indeed.

8) in 3-27 and 3-30 it looks like the quaternion is multiplying a bracket with 3-vectors. I believe that the rotation y quaternion should be applied. Maybe it is only necessary to refine or define the notation, however, the current state is misleading.

9)I am unsure wheteher the state of the UAV is vector or matrix:
a) Is Xk from the first equation (unnubered, why?) a 7x3matrix? The description sugests so.
b) Is x defined on p7, ln 240 a vector or matrix. Since all its elements are column vectors it looks like it is 5x3 matrix after the transposition. Nontheless, it should be vector for the use in EKF.

10) I think that justification or clarification of term "global ArUco" should be provided. According to figures the system works in works insome kind of local tangential plane, where xy axes are perpendicular to the coal deposit walls and z points up. I would expect that "global" navigation implies some connection to an ECEF system.

11) In Section 4 the gyroscope and accelerometer specifications are given as initial deviation and operation deviation terms ar used. I believe that the terms like initial bias , in-run bias stability, angular random walk or velocity random walk are used usually in the IMU specifications.

12) p.11, ln.383: What is the length of the ArUco marker? is it the length of the side of the marker rectangle?

13) There is a repeating sentence in the conclusion on lines 518 and 519. "The data set is used to thoroughly ..."

14) I am missing a comparison with a groud-truth trajectory. Would it be possible to perform a similar test in the open-sky conditions, where GNSS-RTK could be used as a trajectory reference? It would definitely allow a quantitative comparison of the algorithms, which would be very helpful and would improve the overal quality and significance of the article.

Reviewer 2 Report

 this paper proposes a Lidar navigation system based on global ArUco, widely used in large-scale known GNSS-denied scenarios for UAVs. The system jointly optimizes the Lidar, inertial measurement unit, and global ArUco information by factor graph and outputs the pose in the real-world frame. 

In the experiments, computational load of the proposed method needs to be analyzed in detail.

The actual flight trajectory of the UAV is not plotted. Thus, I cannot see the error in the localization process.

LOAM and LIO-SAM are compared in the experiments. References are required for LOAM and LIO-SAM.

The experiment is performed only once. Monte-carlo simulations are required to verify the localization performance robustly.

What is "determination of loopback"? The definition of "loopback" is not clear to me.

Why did you use EKF? You can use UKF or Particle filter for handling nonlinear systems.

Why do you want to localize the UAV in known indoor environments? The application scenario of this paper is not clear to me. 

If you want to localize the UAV in known indoor environments, many methods can be used.  You mention that the UWB positioning preset hardware cost is high, the blind area is large, the maintenance is complex, and the accuracy is easily affected by dust and metals in the environment. However, the advantage of your method compared to UWB positioning is not clear to me. Your method uses camera, which is easily affected by dust in the environment. 

You can simply set up multiple cameras at known positions in the workspace. The UAV image of each camera can be merged to localize the UAV. This is feasible, since the UAV image can be used for Angle-of-Arrival for UAV positioning.  In other words, AOA localization can be used for vision-based localization of the UAV in a known workspace.